


# A Semi-Automated Algorithm to Quantify Scarp Morphology (SPARTA): Application to Normal Faults in Southern Malawi

Michael Hodge[1,5], Juliet Biggs[2,5], Åke Fagereng[1,5], Austin Elliott[3,5], Hassan Mdala[4], and Felix Mphepo[4]

[1]School of Earth and Ocean Sciences, Cardiff University, Cardiff, UK
[2]School of Earth Sciences, University of Bristol, Bristol, UK
[3]Department of Earth Sciences, University of Oxford, Oxford, UK
[4]Geological Survey Department, Mzuzu Regional Office, Malawi
[5]Centre for Observation and Modelling of Earthquakes, Volcanoes and Tectonics (COMET)

**Correspondence:** Michael Hodge (hodgems@cardiff.ac.uk)

**Abstract.** Along-strike variation in scarp morphology reflects differences in a fault's geomorphic and structural development and can thus indicate fault rupture history as well as mechanical segmentation. Parameters that define scarp morphology (height, width, slope) are typically measured or calculated manually. The time-consuming manual approach reduces the density and objectivity of measurements, and can lead to oversight of small-scale morphological variations that occur at a resolution

impractical to capture. Furthermore, inconsistencies in the manual approach may also lead to unknown discrepancies and uncertainties between, and also within, individual fault scarp studies. Here, we aim to improve the efficiency, transparency and uniformity of calculating scarp morphological parameters by developing a semi-automated Scarp PARameTer Algorithm (SPARTA). We compare our findings against a traditional, manual analysis and assess the performance of the algorithm using a range of elevation model resolutions. We then apply our new algorithm to a 12 m resolution DEM for four southern Malawi

fault scarps, located at the southern end of the East African Rift System: the Bilila-Mtakataka fault and three previously unreported scarps - Thyolo, Muona and Malombe. All but Muona exhibit first-order structural segmentation at their surface, and by using a 5 m resolution DEM derived from high-resolution stereo satellite imagery for the Bilila-Mtakataka fault scarp, we are able to quantify secondary structural segmentation. Our scarp height calculations from all four fault scarps suggests that if each scarp was formed by a single, complete rupture, the slip-length ratio for each fault exceeds the maximum typical

value observed empirically in historical normal faulting earthquakes around the world, implying that their structural histories are more complex. The distribution of vertical displacement at the surface implies the structural segments of both the BMF and Thyolo fault have merged via rupture of discrete faults (hard-links) through several earthquake cycles, and the segments of the Malombe fault have connected via distributed deformation zones (soft-links). For all faults studied here, the length of earthquake ruptures may therefore exceed the constitutive length of each segment. Thus, our findings shed new light on the

seismic hazard in southern Malawi, indicating evidence for a number of large ($M_W$ 7 - 8) prehistoric earthquakes, as well as providing a new semi-automated methodology (SPARTA) for calculating scarp morphological parameters, which can be used on other fault scarps to infer structural development.





## 1 Introduction

Earthquake ruptures that break the Earth's surface result in the offset of landforms such as river channels, alluvial fans and other geomorphic features (e.g., Hetzel et al., 2002; Zhang and Thurber, 2003), and create fault scarps that are themselves indicative of the style and magnitude of the earthquake event (Wallace, 1977). By measuring the offsets of landforms and fault

scarps, the earthquake-induced surface displacement along the fault can be determined, which can provide information about the rupture and slip history on the fault (e.g., Wallace, 1968; Sieh, 1978; Zielke et al., 2012; Ren et al., 2016), and be used to identify structural segmentation (e.g., Watterson, 1986; Giba et al., 2012; Manighetti et al., 2015) and the presence of linking structures (e.g., Soliva and Benedicto, 2004; Nicol et al., 2010). For faults whose component segments remain unconnected at the surface, the distribution of displacement along a fault can also provide clues to the future structural development (e.g.,

Walsh and Watterson, 1988; Cowie and Scholz, 1992; Dawers et al., 1993; Dawers and Anders, 1995; Peacock, 2002) by indicating soft-linkages between segments (Willemse et al., 1996; Hilley et al., 2001), such as relay ramps. Over time, these segments may hard-link , i.e. a connection establishes between segments via rupture of discrete fault planes (e.g., Childs et al., 2017). Thus, using a combination of the displacement distribution along a fault and the inter-segment zone geometry, we can understand what linkage might exist at depth (e.g., Crider and Pollard, 1998). The importance of distinguishing between these

two types of linkages are that the physical connection of a hard-link may permit through-going earthquake rupture, and will also increase fluid transport, with implications for reservoirs.

In the past, calculating the displacement across a fault scarp was performed by local field surveys or using Global Positioning System (GPS) devices (e.g., Bucknam and Anderson, 1979; Andrews and Hanks, 1985; Cowie and Scholz, 1992; Gillespie et al., 1992; Cartwright et al., 1995; Avouac, 1993; Delvaux et al., 2012). However, recent advances in remote sensing have

meant that highly accurate and precise vertical displacements can be measured using satellite images and digital elevation models (DEM) (e.g., Westoby et al., 2012; Bemis et al., 2014; Johri et al., 2014; Zhou et al., 2015; Roux-mallouf et al., 2016; Talebian et al., 2016). Depending on resolution, DEMs are categorised as low ($\geq$ 30 m), intermediate ($\sim$ 10 m) or high resolution ($\leq$ 5 m). There is a trade-off between DEM resolution and cost as launching satellites and acquiring (tasking) images is expensive. High resolution DEMs generated by the newest satellites are expensive, exacerbated by large minimum

coverage areas (typically $\sim$ 100 km$^2$). Furthermore, generating a DEM using high resolution satellite images may require pre-processing steps including pan-sharpening, and stereo-alignment. As a satellite programme becomes discontinued, satellite images and DEMs are often released for scientific use at no cost (e.g., the SPOT Historical archive, SRTM). These products require limited, to no, post-processing.

With the current drive toward acquisition of high resolution DEMs for paleoseismological studies (e.g., Zhou et al., 2015;

Roux-mallouf et al., 2016; Talebian et al., 2016), two scientific questions arise: (1) what DEM resolution is required to successfully locate, calculate and accurately analyse the significant changes in displacement along a fault scarp; and (2) does our interpretation of the distribution of displacement scale with DEM resolution (i.e. how much more are we able to infer using an expensive, high resolution DEM compared to a free, lower resolution alternative)?



Despite the advances in satellite and computing technology, and thus the resolution of DEMs, calculating the vertical displacement along a scarp is largely a manual process that has remained consistent over several decades (e.g., Wallace, 1977; Bucknam and Anderson, 1979; Avouac, 1993; Wu and Bruhn, 1994; Ganas et al., 2005; Walker et al., 2015). Scarp height is typically used as a proxy for minimum vertical displacement (e.g., Morewood and Roberts, 2001), and is calculated by first

identifying the fault scarp from an elevation profile by manually picking the crest (top) and base (bottom) of the fault scarp. However, picking the fault scarp location manually can be unrepeatable for intermediate or low resolution DEMs, with measurements showing a variation in picked scarp location for multiple, independent analyses on the same profiles (e.g., Hodge et al., 2018a). Manually processing data can also be subject to human bias; one person's definition of the crest and base of a fault scarp may be different to another person's (Middleton et al., 2016). These inconsistencies ultimately lead to errors within

the scarp height calculations and are a contributing factor for the scatter observed in global maximum displacement-length profiles (Gillespie et al., 1992) and along-strike displacement profiles (Zielke et al., 2015).

In this paper, we develop an algorithm that calculates the parameters (height, width and slope) of a fault scarp from a scarp elevation profile: Scarp PARameTer Algorithm (SPARTA). Using the scarp height as a proxy for vertical displacement (e.g., Morewood and Roberts, 2001), a displacement profile can be created by calculating scarp height at intervals along a fault

scarp. This displacement profile can then be used to infer fault structural segmentation and the existence of secondary linking faults (e.g., Crone and Haller, 1991; Cartwright et al., 1995; Dawers and Anders, 1995; Childs et al., 1996; Giba et al., 2012). Automating the morphological calculations will allow a greater number of measurements to be taken along a fault scarp than feasible with ground based methods, improving the understanding of fault behaviour and segmentation (e.g., Zielke et al., 2012, 2015; Trudgill and Cartwright, 1994; Cartwright et al., 1995; Manighetti et al., 2015). Our goal is to develop an algorithm that

is open-source and able to run on a personal computer. We test the performance of the algorithm using a number of synthetic and real fault scarps, for a variety of DEM resolutions.

Algorithms for relative dating of fault scarps, by performing best fit calculations to a scarp-like template, have already been attempted (e.g., Gallant and Hutchinson, 1997; Hilley et al., 2010; Stewart et al., 2017); however, these methods may falsely identify other geomorphic features as fault scarps, and require a very high resolution DEM, usually obtained using

LiDAR. These autonomous algorithms therefore still require post-processing, manual quality checks. In addition, Shaw and Lin (1993) developed an algorithm to identify fault scarps by measuring topographic curvature within a moving window, however, their method only distinguishes between different relative scarp heights, rather than provide a quantitative measurement of scarp height. The algorithm created here (SPARTA) will be developed to be used for a range of DEM resolutions, where the performance between resolutions is tested in this study.

**2 Normal faults in southern Malawi**

The Malawi Rift System (MRS) exists at the southern end of the East African Rift System, extending 900 km from the Rungwe province in the north to the Urema graben in the south (Specht and Rosendahl, 1989; Ebinger et al., 1987). At the northern end of the rift system is the Mbeya box, which is a triple junction between the Somalian, Victoria and Rovuma plates (Ebinger





et al., 1989). Rift development commenced around *ca.* 8 Ma (Ebinger et al., 1989) with the formation of half-graben units bounded by fairly north-south striking normal faults and propagated from the north (Ring et al., 1992). Kinematic models of plate motion suggest maximum average extension rates across the Malawi rift of $\sim$ 3 mm per year, decreasing southwards to less than 2 mm per year (Jackson and Blenkinsop, 1997; Saria et al., 2014; Jestin et al., 1994; Stamps et al., 2008). Border fault

systems exist with a predominantly north-south trend at the edges of Lake Malawi and alternate sides of the lake at around 100 km intervals (Rosendahl et al., 1986; Ebinger et al., 1987).

In the southern MRS, the Bilila-Mtakataka fault (BMF) scarp breaks the surface along almost its entire length, a distance of $\sim$ 110 km (Jackson and Blenkinsop, 1997). The morphology and geometry of the scarp, however, varies along strike (Hodge et al., 2018a) and is therefore typical of a large, structurally segmented normal fault (e.g., Schwartz and Coppersmith, 1984;

Wesnousky, 1986; Peacock and Sanderson, 1991). The fault is suggested to comprise six, major (first-order) segments, varying in length from 13 km to 38 km, and the distribution of scarp height is of two symmetrical bell-shaped profiles separated by the Citsulo segment (Hodge et al., 2018a). The relatively coarse measurement resolution of the former studies along the BMF have meant that secondary (second-order) segments were unable to be identified or characterised, i.e. subordinate segments that have a length of the same order of magnitude as the major segment they exist within (Manighetti et al., 2015). Although

secondary segments are unlikely to contain gaps of sufficient distance (typically inferred to be $\geq$ 6 km) to perturb rupture propagation (e.g., Gupta and Scholz, 2000; Biasi and Wesnousky, 2016), their existence may provide evidence for the earliest structural development of the fault (e.g., Manighetti et al., 2007). Furthermore, understanding structural segmentation is crucial in estimating earthquake magnitude, as faults segments may rupture individually, consecutively or continuously (e.g., Anderson et al., 2017; Hodge et al., 2015).

The latest morphological analysis also concludes that there may be a gap in the BMF scarp across the Citsulo segment (Hodge et al., 2018a). This discontinuity extends for a maximum length of $\sim$ 10 km. A break in continuity of this length may be sufficient to perturb rupture propagation (Biasi and Wesnousky, 2016) and prevent hard-linkage along a normal fault (Hodge et al., 2018b). A reduced maximum rupture length would reduce the maximum expected earthquake magnitude (Wells and Coppersmith, 1994) and also the earthquake repeat time (Hodge et al., 2015). Therefore, in order to conclude whether the

fault scarp is discontinuous across the Citsulo segment, and the existence of secondary segments and associated linking faults, a higher resolution DEM and a greater number of scarp profiles is required.

Although the Bilila-Mtakataka fault provides an ideal case study of a large, continental normal fault, in order to understand whether it is unique or representative of early-stage rift faulting, we extend our research to other fault scarps within the southern, amagmatic Malawi Rift System (MRS). We investigate three additional faults in the southern MRS identified during fieldwork,

which have previously unreported scarps, the Malombe, Thyolo and Muona faults. The Malombe fault is a north-south striking, east-dipping normal fault located $\sim$ 40 km east of the Bilila-Mtakataka fault, on the edge of Lake Malombe; the fault scarp contains at least two major gaps in its surface expression (Fig. 1c). Lithology varies considerably along the fault length, alternating between felsic and mafic paragneisses with fingers of calc-silicate granulite that intersect the scarp (Manyozo et al., 1972). The Thyolo and Muona faults, south of the Bilila-Mtakataka fault, are two overlapping northwest-southeast striking,

southwest-dipping parallel normal fault scarps separated by an offset of $\sim$ 5 km (Fig. 1d). The lithology of the scarp footwall





**Figure 1.** a) Map of faults in southern Malawi; those used in this study are coloured black. Tick marks show the dip direction. Lower left corner shows the plate motion (PM; $86° \pm 5°$) from Saria et al. (2014) and local minimum horizontal stress ($SH_{min}$; $62°$) from Delvaux and Barth (2010). Panels b to d are geological maps of: b) The Bilila-Mtakataka fault (BMF) showing the coverage of the Pleiades satellite imagery (taken from Walshaw, 1965; Dawson and Kirkpatrick, 1968); c) The Malombe faults (MAF): northern Malombe fault (NMAF); central Malombe fault (CMAF); and southern Malombe fault (SMAF) (taken from Manyozo et al., 1972); and d) The Thyolo fault (TOF) and Muona fault (MOF) (taken from Habgood et al., 1973).

is very homogeneous at the regional scale, mapped as mafic paragneiss along its entire length (Habgood et al., 1973). To infer the distribution of scarp height, structural segmentation and linkage structures along the Malombe, Thyolo and Muona fault scarps, we develop an algorithm to calculate profiles of the height and width of the scarp. We then compare our findings for these newly studied faults with the Bilila-Mtakataka fault, and assess their morphology and structural development. We also

5  calculate the slip-length ratio for each fault and compare against typical values for normal faults (Scholz, 2002).





## 2.1 Scarp algorithm

### 2.1.1 Scarp identification

For a given profile perpendicular to the local scarp trend, the first step in calculating the scarp's morphological parameters (height, width and slope) is to identify the crest and base of the scarp. Fig. 2a-c shows three profiles from the Bilila-Mtakataka

fault scarp taken using the Pleiades 50 cm DEM. The black line is the elevation data extracted from the DEM, the red line the change in elevation per unit distance $\partial z/\partial X$ (i.e. slope, $\theta$), and the blue circles are the derivative of slope $\partial^2 z/\partial X^2$ ($\phi$). Each of the three profiles is characteristic of a different challenge associated with picking the fault scarp manually. The quality of the profile is determined by the signal-to-noise ratio, whereby a profile with a clear scarp and little background noise has a high signal-to-noise ratio. Profile A has a high signal-to-noise ratio, and a large, wide scarp; however, the gradient of the scarp is

not constant, leading to large slope derivative values (Fig. 2a). Profile B has a low signal-to-noise ratio, caused by vegetation or other topographical features; this noise creates local variability in slope $\theta$, yet the gradient on the scarp itself is fairly constant (Fig. 2b). Profile C has a low signal-to-noise ratio, the scarp width is small and the magnitude of the change in slope at the fault scarp is not large; it is therefore difficult to accurately identify the scarp from the footwall topography (Fig. 2c). Furthermore, Profile C's morphology makes picking a fault scarp even more challenging when using a lower resolution DEM.

For each profile in fig. 2, grey triangles denote a manual pick of the crest and base of the fault scarp. We consider the basic assumption that the fault scarp represents the approximate position of the fault. A linear regression (least squares method) is then applied to the upper original and lower original surfaces. The best-fitting lines for the upper and lower original surfaces (grey dotted lines) are then extrapolated to the point of maximum slope ($\theta_{max}$) on the identified fault scarp. The scarp height $H$ is then taken as the elevation difference between the regression lines at this point, the gradient of the best-fit line through

the fault scarp is the scarp slope $\alpha$, and the horizontal distance between fault scarp crest and base is the scarp width $W$.

Our algorithm picks the crest and base of the fault scarp based on the first and last values of the scarp profile that satisfy *a priori* threshold values of slope ($\theta_T$) and the derivative of slope ($\phi_T$). For the algorithm to calculate accurate values for scarp height, width and slope, the thresholds need to be appropriate for the scarp's morphology, i.e. for gently dipping fault scarps the slope threshold should also be of a gentle angle. Two examples for slope threshold are shown for the profiles in fig. 2, one

where the slope threshold is set to 20° (pink triangles) and one where it is 40° (blue triangles). For all profiles, neither threshold value performs well at automatically identifying a fault scarp equivalent to the one that was determined manually. The reason for the poor algorithm performance is the low signal-to-noise ratio, whereby noise within the original surfaces may lead to the misidentification of the fault scarp by the algorithm. For example, in fig. 2a and 2c, the algorithm fails to accurately identify the base of the fault scarp due to noise in the lower original surface. For all examples, the crest of the fault scarp is misidentified

by the algorithm due to noise within the upper original surface. Furthermore, as the values of $\phi$ have a high amplitude than $\theta$, the algorithm is more sensitive to the slope derivative threshold than the slope threshold. To enhance the signal-to-noise ratio of the elevation profiles, we apply and test a range of digital filters.





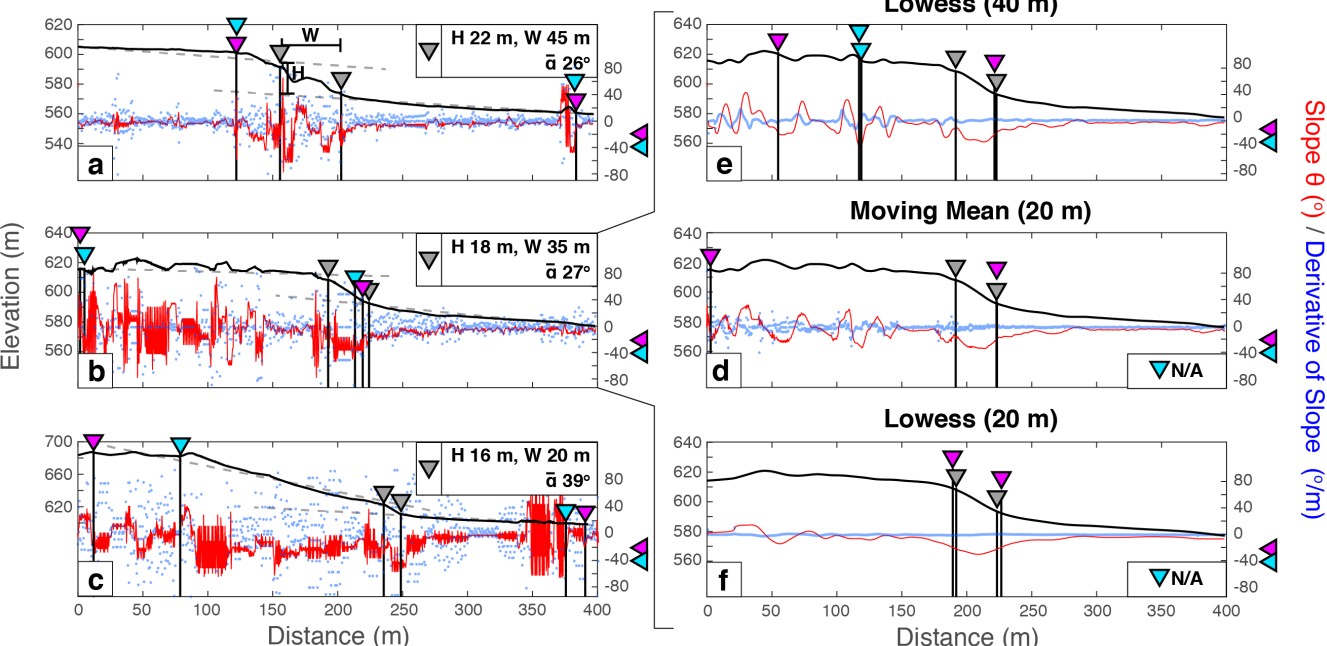

**Figure 2.** Panels a to c) Three profiles across the Bilila-Mtakataka fault scarp using the Pleiades 50 cm DEM. Each is characteristic of the different challenges associated with picking the fault scarp manually and using an algorithm. Profile A has a high signal-to-noise ratio but contains noise on the fault scarp. Profile B and C have a lower signal-to-noise ratio, and Profile C has a scarp that is difficult to accurately identify as the magnitude of the change in slope at the fault scarp is not large. Using Profile B, three different digital filters and/or bin widths were applied: d) Lowess (bin width 20 m); e) Moving Mean (bin width 20 m); and f) Lowess (bin width 40 m). The black line is the elevation profile, the red line is the slope ($\theta$) profile and blue circles denote the derivative of slope ($\phi$). Grey triangles show the location of the crest and base of the fault scarp based on a manual pick. Pink and blue triangles denote the algorithms pick of the crest and base based on a slope threshold of 20° (pink) and 40° (blue), respectively.

### 2.1.2 Filtering

Here, we test the suitability of four digital filters (Moving Mean, Moving Media, Savitzky-Golay and Lowess) in improving the signal-to-noise ratio of the scarp profiles and improving the accuracy with which morphological parameters such as height and width can be extracted by automated processing. Each filter uses a moving window over a specified bin width, which must

5   be an odd integer. The moving window is incrementally shifted along the profile for each datapoint.

For the Moving Mean and Moving Median filters, we use the rolling mean algorithm from the *pandas* Python module and the moving median algorithm from the *SciPy* Python module, repectively. Both filters are commonly used signal-processing algorithms because they are the easiest and fastest digital filters to understand and use. In image processing, the Median filter is usually the preferred digital filter because it better represents the average. This is because an individual unrepresentative value in the window will not affect the median value as significantly as it affects the mean. However, the Median filter also preserves





sharp edges and therefore may lead to step-like features, which could cause steep slope artefacts in our profiles. The Savitzky-Golay filter is based on a local least-squares polynomial approximation (Savitzky and Golay, 1964); it is less aggressive than simple moving filters and is therefore better at preserving data features such as peak height and width. The Lowess filter uses a non-parametric regression method and requires larger sample sizes than the other filters (Cleveland, 1981). The Lowess filter

can be performed iteratively, but since it requires much more computational power than the other filter methods, we apply a single pass over the data.

Fig. 2d-f shows the results of applying a digital filter to Profile B (Fig. 2b). This profile was chosen because of the extensive noise within the upper original surface. Such noise is typical for fault scarp profiles, as topographic features from previous deformation events, valleys and dense vegetation are common. The elevation data was filtered using the following parameters:

d) Lowess (bin width 20 m); e) Moving Mean (bin width 20 m); and f) Lowess (bin width 40 m). Filter parameters for Profiles D and E were chosen as a comparison between two different filter methods using the same bin width, whilst parameters for Profiles D and F were chosen for a comparison between different bin widths for the same filter method.

The Lowess filter smoothes the elevation, and subsequently the profiles of slope $\theta$ and slope derivative $\phi$, more than the Moving Mean filter. As expected, a larger bin width smoothes the data more than a smaller bin width. By smoothing the data,

the relative amplitude of $\phi$ becomes smaller than that of $\theta$, meaning that the algorithm becomes less sensitive to the slope derivative threshold than the slope threshold. For the same bin width (20 m), the algorithm using the Lowess filter estimates the scarp location more accurately than the Moving Mean for this profile, as the latter fails to significantly reduce the noise within the upper original surface; however, for both filters, the algorithm still falsely identifies the crest of the fault scarp using a slope threshold of 20° or 40°. The algorithm, using a slope threshold of 20°, performs reasonably well once the profile has

been filtered using the Lowess filter and a bin width of 40 m (Fig. 2f), for this example.

## 2.2   Assessing algorithm performance

We assess the performance of our algorithm by testing it on various scarp profiles. Performance is assessed by defining a misfit value for scarp height ($H_m$), width ($W_m$) and slope ($\alpha_m$) as the difference between ground-truthed ($H_g$, $W_g$, $\alpha_g$) and algorithm calculated ($H_c$, $W_c$, $\alpha_c$) scarp parameters - based on the selected *a priori* parameters $b, \theta_T, \phi_T$ and filter method - for

each profile. Misfit values can be positive or negative. This approach relies on the assumption that the ground-truthed value is correct, and is the value that we want the algorithm to calculate. One approach, as shown above, is to use a manual analysis to calculate the ground-truthed values. For example, for Profile F, the crest and base were both identified by the algorithm within 5 m of the manual pick, leading to a height misfit of less than 1 m, a width misfit of less than 6 m, and a slope misfit smaller than 1° (Fig. 2f). Another way to test algorithm performance is to generate a synthetic fault scarp profile where the ground-truthed

values are the known synthetic scarp parameters.

Although the ultimate goal is to design an algorithm to calculate scarp parameters for real fault scarps, the creation of a synthetic catalogue will allow us to robustly test the algorithm, and the relationship between filter and threshold parameters, using a large number of scarp profiles. This would not be feasible using the manual process. Therefore, the algorithm is run iteratively on a number ($n$) of synthetic profiles, using a range of *a priori* filter and threshold values. Average height ($\bar{H}_m$),





width ($\bar{W}_m$) and slope ($\bar{\alpha}_m$) misfit values are then calculated using the mean of individual misfit values from the profiles (equations 1 to 3). The total number of profiles where a fault scarp is identified by the algorithm is given as the count $C$. The total misfit value, $\varepsilon$, is then calculated using equation 4; all algorithm runs where number of fault scarps identified is fewer than 50% are removed. Although calculating the correct scarp height is the most important element of our algorithm, an equal

weight is applied to all scarp parameters because all contribute to how well the scarp is identified. The smallest $\varepsilon$ value is then used to denote the best performing set of filter and threshold parameters.

$$\bar{H}_m = \frac{1}{n} \sum_{i=1}^{n} H_{c(i)} - H_{g(i)} \tag{1}$$

$$\bar{W}_m = \frac{1}{n} \sum_{i=1}^{n} W_{c(i)} - W_{g(i)} \tag{2}$$

$$\bar{\alpha}_m = \frac{1}{n} \sum_{i=1}^{n} \alpha_{c(i)} - \alpha_{g(i)} \tag{3}$$

$$\varepsilon = \frac{|\bar{H}_m| + |\bar{W}_m| + |\bar{\alpha}_m|}{C/n} \qquad\qquad , \text{for } C \geq 0.5n \tag{4}$$

## 3  Synthetic tests

### 3.1  Synthetic catalogue

In order to test the possible combinations of filtering, bin sizes etc using a Monte Carlo approach, we construct two synthetic catalogues, noise-free and noisy, each comprising 1,000 fault scarp profiles.

The parameters used in the construction of both catalogues are: the location of the scarp crest along the profile ($x_s$); the slope of the upper original surface ($\beta_u$); and the slope of the lower original surface ($\beta_l$, Table 1, Fig. 3a). Profile length $x$ and resolution $r$ are constants set to 400 m and 1 m, respectively. Parameters $\beta_u$ and $\beta_l$ could be omitted if the synthetic catalogue is used to mimic an environment where fault scarps offset flat surfaces (e.g., Borah Peak fault scarp, Idaho; Ward and Barrientos, 1986), and included for regions where fault scarps offset sloped surfaces (e.g., Mangola fault scarp, Central

Apennines; Tucker et al., 2011). A down-dip, normal sense of displacement parallel to the scarp is then imposed and $Z$ and $X$ are defined as the vertical (throw) and horizontal (heave) components of this displacement. The synthetic fault scarp width $W_g$ therefore equals the horizontal displacement $X$ and scarp slope $\alpha_g$ equals $\tan(Z/X)$. The height of the synthetic fault scarp $H_g$ is then calculated using equation 5. The larger the values of $\beta_u$ and $\beta_l$, the larger the difference between measured throw and actual throw, $H_g$ and $Z$ (Fig. 3b).

$$H_g = Z - \frac{X}{2}(\tan \beta_u + \tan \beta_l) \tag{5}$$





The noisy catalogue includes noise in the form of vegetation, hills and ditches, as well as scarp degradation by diffusion (Table 1; Fig. 3c). A random number of these noisy features are then placed at a random location along the profile. The shape of these noisy features is a negative parabola between $a$ and $b$, created using equation 6, where $a$ is the first root at the random location and $b$ is the second root at a horizontal distance from the first root equating to the feature width, with a height $\frac{-kb^2}{4}$.

$$y = -k(x-a)(x-a-b) \tag{6}$$

Diffusion is applied in a Monte Carlo approach by using equation 7 for a diffusion constant $\kappa$ and time $t$, resulting in erosion of material from the upper portion of the scarp and deposition at the base (Fig. 3c). Diffusion can be included for environments where hillslopes are mantled with a continuous soil cover (i.e. transport-limited) and excluded for those with extensive areas of bare bedrock (i.e. weathering-limited) (e.g., Tucker et al., 2011; Bubeck et al., 2015; Boncio et al., 2016). Early studies of scarp

10 degradation suggested that the value of $\kappa$ should typically be between 0.5 and 1.5 m$^2$/kyr (e.g., Hanks et al., 1984; Andrews and Hanks, 1985; Arrowsmith et al., 1996); however, recent studies from Mongolia (Carretier et al., 2002), the Gulf of Corinth (Kokkalas and Koukouvelas, 2005) and the upper Rhine valley (Nivière and Marquis, 2000) have suggested $\kappa$ values in the range of 3 to 10 m$^2$/kyr. Locally on scarps in the Gulf of Corinth, $\kappa$ has been measured to be as low as 0.2 m$^2$/kyr (Kokkalas and Koukouvelas, 2005), however, errors in calculations can be as large as 0.5 m$^2$/kyr. Here, we set algorithm limits to 0.5 and

15 10 m$^2$/kyr.

$$\frac{dh}{dt} = \kappa \cdot \frac{d^2h}{dx^2} \tag{7}$$

| Algorithm Parameters | | | This Study | |
|---|---|---|---|---|
| Parameter | Symbol | Unit | Minimum value | Maximum value |
| *All Catalogue Parameters* | | | | |
| Profile length | $x$ | metres (m) | 400 | - |
| Scarp location | $x_s$ | metres (m) | 100 | 300 |
| Vertical offset | $Z$ | metres (m) | 2 | 50 |
| Horizontal offset | $X$ | metres (m) | 2 | 100 |
| Upper slope | $\beta_u$ | degrees (°) | 5 | 0 |
| Lower slope | $\beta_l$ | degrees (°) | 5 | 0 |
| *Additional Noisy Catalogue Parameters* | | | | |
| Diffusion constant | $\kappa$ | m$^2$/kyr | 0.5 | 10 |
| Chronological age | $t$ | kyr | 0 | 50 |
| Vegetation number | $v_n$ | dimensionless | 0 | 20 |
| Vegetation height | $v_H$ | metres (m) | 1 | 3 |
| Vegetation width | $v_W$ | metres (m) | 1 | 3 |
| Hill number | $h_n$ | dimensionless | 0 | 3 |
| Hill height | $h_H$ | metres (m) | 3 | 10 |
| Hill width | $h_W$ | metres (m) | 8 | 15 |
| Ditch number | $d_n$ | dimensionless | 0 | 3 |
| Ditch depth | $d_H$ | metres (m) | 3 | 10 |
| Ditch width | $d_W$ | metres (m) | 8 | 15 |

**Table 1.** Parameters used in creating the synthetic catalogues.





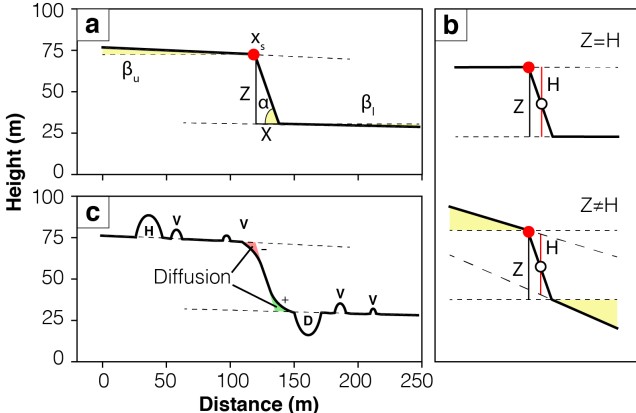

**Figure 3.** An example of a synthetic catalogue fault scarp. a) Visual description of the parameters in Table 1 used in the noise-free synthetic catalogue. b) The difference between vertical displacement $Z$ and synthetic profile scarp height $H_g$, resulting from sloping original surfaces. c) The additional noisy synthetic catalogue parameters (H = hill; V = vegetation; and D = ditch) and diffusion (red - erosion, green - deposition).

## 3.2 Individual profiles

We test the performance of the algorithm by comparing ground-truthed synthetic scarp values to scarp parameter values calculated by the algorithm. The synthetic catalogue input values are shown in Table 1. All filters from Section 2.1.2 were tested, using a bin width between 9 and 99 m, increasing in increments of 10 m. We vary slope threshold, $\theta_T$, between 1° and 41°, in increments of 10°, and fix the slope derivative threshold, $\phi_T$, to 5°/m.

Fig. 4a shows five examples with various morphologies from the noise-free synthetic catalogue: P1) randomly selected; P2) small scarp height; P3) steep, large scarp; P4) gently dipping, parallel original surfaces; and P5) non-parallel original surfaces. The algorithm was tested using all combinations of filter methods, bin widths and slope thresholds. For each profile, misfit values were calculated (Fig. 4a). For scarp width and slope misfit for synthetic catalogues, see the supporting information. For all examples, the algorithm was able to identify a fault scarp and report scarp height with a misfit of less than 2.5 m (5% - 60% of the scarp height for some combination of parameters); however, for Profile 2, the algorithm was unable to identify a fault scarp when the bin width was greater than 30 m. In this case, the filter was too aggressive and over-smoothed the scarp, such that no clear break in slope was detectable. Detectability of the scarp slope is a function of resolution, scarps may not be identified if the bin width is three times the scarp width and height, and the misfit values are greater for bin widths twice the scarp width and/or height.

To illustrate the process, we chose three examples from the noisy synthetic catalogue based on their signal-to-noise ratio and diffusion parameters (Fig. 4b). Profile 6 includes lots of vegetation but no hills or ditches (moderate signal-to-noise ratio), nor any scarp diffusion. Profile 7 includes hills, ditches and scarp diffusion, but no vegetation (high signal-to-noise ratio). Profile 8





includes vegetation, hills and ditches and therefore has the largest amount of noise (low signal-to-noise ratio), and also includes scarp diffusion. For all three profiles, using no filter or the Moving Median filter gave the largest misfit values (Fig. 4b). For scarp width and slope misfit, see the supporting information. The Moving Mean filter provided a small scarp height misfit ($<$ 2.5 m) for Profiles 6 and 7, but produced a larger misfit ($H_m > 7.5$ m) for Profile 8. The Savitzky-Golay and Lowess filters

performed equally well on all profiles, with the former able to identify fault scarps with a slightly larger bin width and steeper slope threshold than the latter.

### 3.3 Exploration of parameter space using synthetic catalogue

For each of the 1,000 profiles in the synthetic catalogues, we test 250 unique combinations of algorithm parameters (filter method, bin width, and slope threshold) and assess their ability to accurately determine the synthetic input parameters. Where

the algorithm is not able to identify a fault scarp, a result is not recorded.

Fig. 5a shows the average misfit values for the noise-free synthetic profiles where the algorithm identified a fault scarp (equations 1 to 3). The best performing bin width and slope threshold depended on the filter method used, but in general a smaller bin width and steeper slope threshold provided smaller misfit values. When not applying a filter, or using the Median filter, the algorithm performed poorly; but using these filters meant the fault scarp was identified in more profiles. For the

Moving Mean, Savitzky-Golay and Lowess filters, a gentle slope threshold ($\theta_T < 11°$) gave large misfit values, but using a steep threshold ($\theta_T \geq 31°$) meant fault scarps were identified in less than 50% of the profiles.

The poor algorithm performance when not using a filter, or using the Moving Median filter, is apparent for the average misfit values using the noisy catalogue (Fig. 5b). On average, the scarp width misfit values are larger than the scarp height misfit values. Whereas scarp height is estimated by linear extrapolation of the original surfaces and is therefore less influenced by

noise and exact position of the fault scarp, scarp width is highly sensitive to the exact location of the fault scarp crest and base picked by the algorithm.

For both synthetic catalogs, the best performing filters were the Savitzky-Golay and Lowess filters, the slope threshold with the smallest total misfit (equation 4) was 21°, and a bin width 50 m or smaller was found to perform better than a larger bin width. Thus these are the optimal filters which we choose to employ in our natural measurements, but we shall undertake

another misfit analysis to identify the best performing bin width and slope threshold.

### 4 Case study example: The Bilila-Mtakataka fault

For the SRTM, TanDEM-X and Pleiades DEMs, hillshade and slope maps were produced in QGIS 2.18 and used to identify the breaks in slope associated with the Bilila-Mtakataka fault, i.e. the scarp. Fig. 6 shows the hillshade image produced by each DEM for an area of the Bilila-Mtakataka fault scarp. The scarp trace was manually picked from each hillshade image and

is shown by a red line. Large-scale changes in scarp trend can be identified using the SRTM DEM (box A, Fig. 6), however, small-scale changes may not be identifiable (boxes B and C, Fig. 6).





**Figure 4.** Scarp height misfit $H_m$ for a) five noise-free synthetic catalogue examples, and b) three noisy synthetic catalogue examples. See the supporting information for scarp width $W_m$ and slope misfit $\alpha_m$ results for the noise-free and noisy catalogues.





**Figure 5.** The average misfit and count for 1,000 a) noise-free and b) noisy synthetic catalogue fault scarps. Grey values denote no fault scarp was identified for all profiles. For resolutions of 5, 10 and 30 m, see the supporting information.



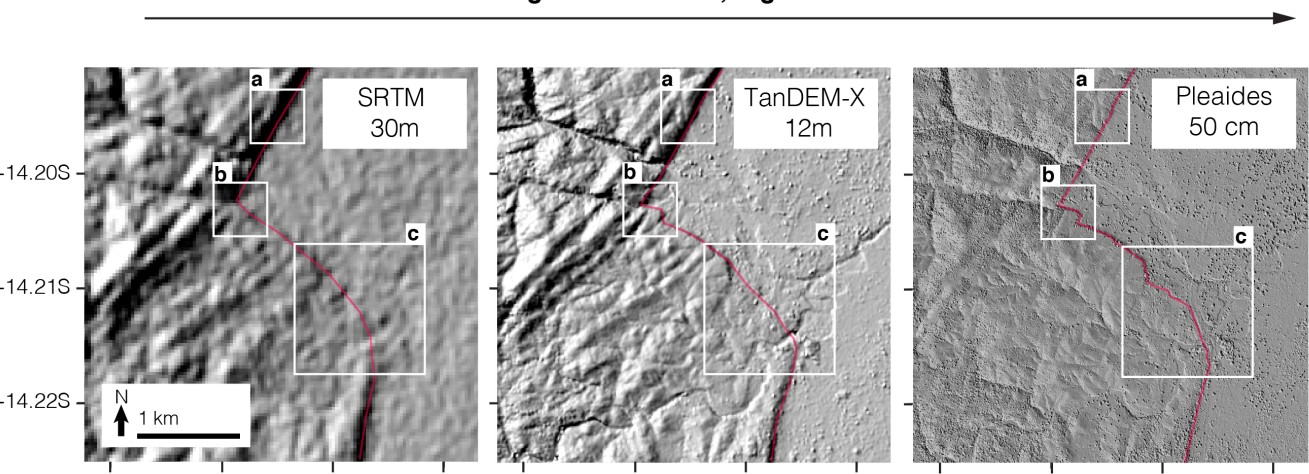

**Figure 6.** Bilila-Mtakataka fault scarp hillshade DEM examples using SRTM 30 m, TanDEM-X 12 m and Pleiades 50 cm DEMs. The red line represents the fault scarp trace picked using each DEM. Box A represents the typical trend of the Bilila-Mtakataka fault scarp, boxes B and C show changes in variation in scarp trend.

We use the station lines toolbox in QGIS to draw profile lines perpendicular to the manually picked fault scarp trace. The total length of the profile $x$ was set to 400 m. To obtain accurate calculations of the scarp's morphological parameters (especially width and slope), profiles need to be taken perpendicular to the scarp trend. Therefore, where the scarp trend varies considerably, such as at the ends of fault segments and at linking structures, failing to account for the small changes in scarp

trend may lead to inaccurate morphological measurements. To prevent the station lines being drawn oblique to the true fault scarp, resulting from small-scale changes in scarp geometry, the distance between nodes (points picked on the fault scarp that when joined represent the scarp trace) should be significantly less than the distance between profiles. Here, we select scarp-perpendicular profiles at intervals of 100 m along the fault scarp trace, and therefore use a nodal distance of $\sim$ 20 m. Therefore, as the resolution of the TanDEM-X DEM is smaller than the nodal distance, we use this to pick the surface trace of

the Bilila-Mtakataka fault scarp.

A total of 913 scarp profiles were extracted from the SRTM, TanDEM-X and Pleiades 5 m DEMs, for $\sim$ 90 km of the Bilila-Mtakataka fault scarp that was covered by the Pleiades DEM, starting $\sim$ 7.4 km from the northern fault end (Fig. 1b). Due to clouds over the fault scarp on the Pleiades optical images, 26 profiles between 94 and 97 km from the northern fault end, were removed. Elevation values were taken along each profile at a spacing equal to the resolution of the DEM (e.g., 5 m

for the Pleiades DEM).





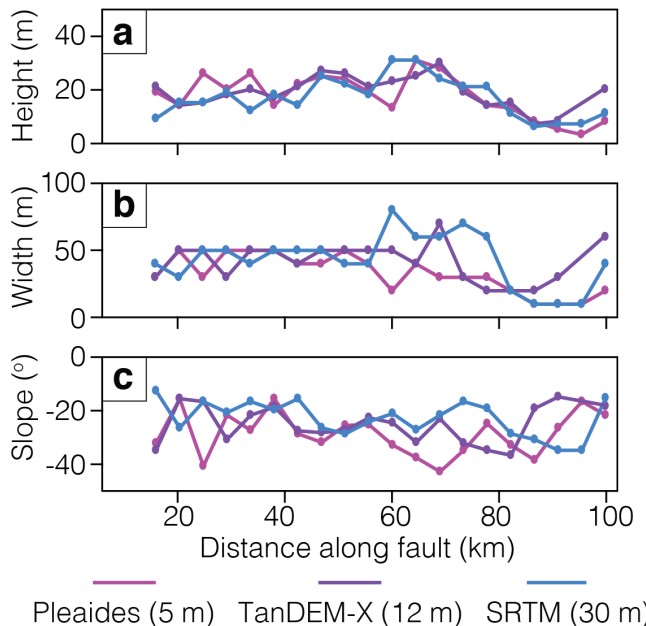

**Figure 7.** Manual Bilila-Mtakataka fault profile for a) height $H$, b) width $W$ and c) slope $\alpha$ taken at $\sim$ 5 km intervals using the Pleiades 5 m, TanDEM-X 12 m and SRTM 30 m DEMs. For tabular results see the supporting information.

## 4.1 Algorithm results (Pleiades 5 m DEM)

To test the algorithm using a range of resolution datasets we first use the Pleiades profiles along the Bilila-Mtakataka fault. A manual analysis is conducted for twenty profiles, taken at increments of $\sim$ 5 km along the Bilila-Mtakataka fault scarp (Fig. 7). A misfit analysis is performed by comparing scarp parameters estimated manually and from the automated analysis.

Based on the algorithm performance in the synthetic tests, we only use the Savitzky-Golay and Lowess filters. The maximum bin width is reduced to 49 m, and slope threshold limits are 11° and 26°, with increments of 5°. We find that the algorithm using the Lowess filter, on average, had smaller misfit values and identified a greater number of fault scarps than using the Savitzky-Golay filter (Fig. 8). As with the synthetic tests, larger bin widths and steeper slope thresholds generated smaller misfit values, especially for scarp width; however, they also identified fewer fault scarps. The algorithm using the Savitzky-

Golay filter gave a large width misfit ($>$ 20 m), except when using the largest bin widths and steepest slope thresholds in the study. Based on the total misfit value, the best results were achieved by the Lowess filter when bin width is 39 m, and a slope and slope derivative thresholds were 21° and 5°/m, respectively. The average misfit values using this algorithm setup were $\bar{H}_m$ = 1.4 m, $\bar{W}_m$ = -6.6 m and $\bar{\alpha}_m$ = -12.6°. These values are specific to this example, and would vary according to DEM resolution, scarp characteristics and location.

Using the best performing parameters the algorithm was able to identify a fault scarp for 79% of the 913 profiles. A histogram of the scarp height, width and slope, as well as the mean and standard deviation ($\sigma$), are shown in fig. 9a (black). The average





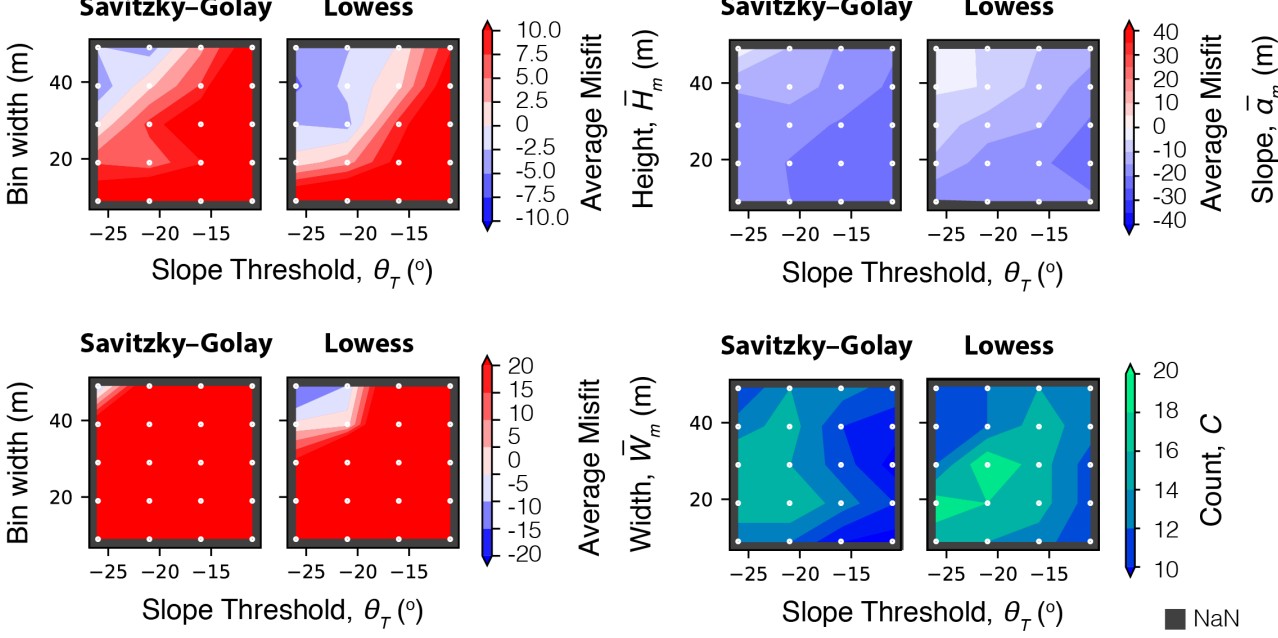

**Figure 8.** Average misfit values between algorithm and manual scarp parameters for twenty Bilila-Mtakataka fault profiles using the Pleiades 5 m DEM.

Bilila-Mtakataka fault scarp height, width and slope were 19 m (± 17 m), 73 m (± 71 m) and 20° (± 12°), respectively. However, as the standard deviation was of the same order of magnitude as the values themselves, this suggests there was a wide spread of results due to natural variability. Furthermore, the extremes exceeded the minimum and maximum values obtained in the manual analysis.

### 5 4.1.1 Resolution analysis

Manual analyses were performed for the twenty chosen profiles along the Bilila-Mtakataka fault scarp using the TanDEM-X and SRTM DEMs, and compared to the Pleiades DEM manual results (Fig. 7). Scarp height esimates between manual analyses differed by a maximum of 18 m, width by up to 60 m and slope by up to 24°, but the average differences were much less: ∼ 4 m, ∼ 13 m and ∼ 8°, respectively. The calculated scarp height and slope were the smallest and most gentle using the SRTM 10 DEM, and tallest and steepest using the Pleiades DEM, likely due to the differing DEM resolutions.

The algorithm was then run for the 913 fault scarps using the TanDEM-X and SRTM DEMs, using the best performing algorithm setup found for the Pleiades analysis. For plots from this resolution analysis, see the supporting information. Although the misfit values were comparable regardless of DEM resolution, the lower the resolution, the fewer the fault scarps that were identified: 69% for TanDEM-X and 64% for SRTM, compared with 79% for Pleiades. The standard deviation of





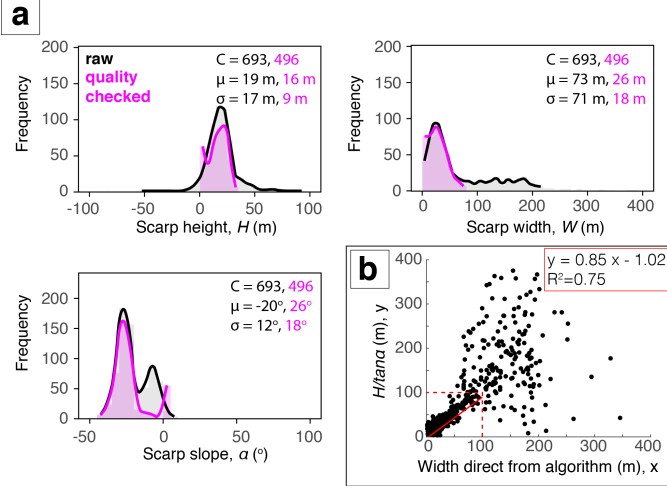

**Figure 9.** a) Histogram of the estimated scarp parameters for the Bilila-Mtakataka fault for all (raw) algorithm estimates (black) and post-quality checked (pink) results. b) A comparison between the scarp width obtained directly from the algorithm against the scarp width calculated using the algorithm's scarp height and slope values ($W_c = H_c / \tan \alpha_c$). A linear regression is applied where width is less than 100 m.

results was smaller for both TanDEM-X and SRTM results than the Pleiades DEM, leading to fewer outliers being removed after the quality check tests were performed. Misfit values were smaller using the higher resolution DEMs. In agreement with the manual analysis, the algorithm scarp parameters were smaller, wider and more gentle on average using the SRTM DEM, but the algorithm was still able to identify scarps with heights less than 5 m.

The average scarp height, width and slope obtained through the algorithm using each DEM were similar. The difference in scarp height between resolutions was smallest between Pleiades and TanDEM-X ($2\sigma < 10$ m) and largest between TanDEM-X and SRTM ($2\sigma \sim 12$ m). The greatest difference in algorithm performance between resolutions was found for scarp width (40 m $> 2\sigma > 20$ m), whereas the difference between scarp slope using each resolution typically was less than $15°$. The difference in scarp height between resolutions did not show any clear along-strike pattern, and was on average less than 5 m. Using a

moving mean, the along-strike changes in scarp parameters between DEMs are similar and match the manual analyses well. For a scarp whose height is comparable to that of the Bilila-Mtakataka's, we find that using a low resolution DEM (i.e., 30 m SRTM) does not profoundly affect the results, however, for smaller scarps and for accurate slope calculations, a high resolution DEM is more appropriate.

## 5    Application to Malombe, Thyolo and Muona faults

We have shown that an automated approach performs well in comparison to a manual analysis for the Bilila-Mtakataka fault scarp. We now apply the algorithm to three further normal fault scarps, the Malombe, Thyolo and Muona fault scarps in southern Malawi (Fig. 1). The Thyolo fault (TOF) and Muona fault (MOF) are two distinct, overlapping fault scarps. As such,



they may be part of the same fault system; however, a physical connection between them is not obvious in the TanDEM-X DEM. The Malombe fault (MAF) is split into three scarps: the northern (NMAF), central (CMAF), and southern (SMAF) scarps. As the algorithm performed comparatively well using TanDEM-X DEM and the Pleaides DEM for the Bilila-Mtakataka fault, we can reliably use TanDEM-X where Pleiades is not available. Therefore for each fault, scarp parameters were calculated using

the algorithm from 400 m long scarp-perpendicular profiles taken using the TanDEM-X DEM. Nodal distance for the manually picked scarp traces is again set to ∼ 20 m and scarp-perpendicular profiles are taken at intervals of 100 m. For each, we select a subsample of twenty-five scarp profiles for a misfit analysis against a manual method (equations 1 to 4), and limit our filter methods to Savitzky-Golay and Lowess.

### 5.1   Scarp morphology of Malombe, Thyolo and Muona faults (TanDEM-X 12 m DEM)

The Thyolo fault scarp is ∼ 70 km long and trends predominantly northwest-southeast (Fig. 1c). Results from the manual analysis indicate that the average height of the TOF scarp is ∼ 18 m, and its average slope is 18°. For results, see the supporting information. The scarp of the parallel Muona fault steps to the right of the Thyolo fault and is shorter, measuring ∼ 28 km long. The faults overlap for a distance of ∼ 10 km and are separated by ∼ 5 km (Fig. 1c). The manual analysis suggests that the MOF scarp is less high (10 m on average) and more gentle (14° on average) than the TOF fault. The scarp width for both faults was

∼ 65 m on average, equivalent to ∼ 5 pixels. The scarp height for both faults increases by up to ∼ 9 m per kilometre toward the overlap zone. Scarp measurements for the TOF within the overlap zone may contain significant errors due to the complex topography within the footwall of the Muona scarp affecting the linear regression of original surfaces. The best performing filter for the TOF was the Lowess filter, whereas the Savitzky-Golay filter performed better for the Muona scarp (Table 2). Both faults required similar slope thresholds, but the TOF required a larger bin width (41 m compared to 29 m). The algorithm

misfit values for the subsampled profiles are shown in Table 2. The algorithm performed less well for the MOF, with an average height misfit of ∼ 12 m, compared to ∼ 6 m for the Thyolo fault.

    The lengths of the Malombe fault scarps are between 16 km and 23 km, with the central scarp being the longest. Again, for results, see the supporting information. All trend approximately north-south with small local changes in scarp trend (Fig. 1d). No hard-linking structures between individual fault scarps were identifiable. Results from the manual analysis show that

the scarps of NMAF and CMAF are morphologically similar, with an average height ∼ 7 m and slope ∼ 9°. The scarp of the SMAF is smaller ∼ 4 m and more gentle ∼ 5°. The widths for all varied on average between 60 m and 80 m. Due to their similar average slopes, the best performing parameters for NMF and CMAF were similar, with the Savitzky-Golay filter preferred (Table 2). The algorithm using the Lowess filter performed best for SMAF, which also performed well using smaller slope threshold and bin width than the fault scarps to the north.

The percentage of fault scarps identified for Thyolo and Malombe profiles was between 50% and 60% (Table 2), yet there were a wide spread of results. To improve the algorithm outcome, first negative scarp heights and positive scarp slopes were removed. Then, as scarp height values for both Thyolo and Malombe were normally distributed, the remaining results were quality checked using a 2σ (95% confidence interval) threshold. Following the quality control, the percentage of scarp profiles





that morphological parameters were measured for was ∼ 30% for all scarps except the southern Malombe fault (13%). This is likely because the small and gentle SMAF scarp may be beyond the detectable limit of profiles using the TanDEM-X DEM.

| Fault Name | Filter | $\theta$ | $b$ | $\bar{H}_m$ (m) | $\bar{W}_m$ (m) | $\bar{\alpha}_m$ (°) | Count, $C$ (%) |
|---|---|---|---|---|---|---|---|
| TOF | LW | 19 | 41 | 6.2 | -1.5 | -0.6 | 60% |
| MOF | SG | 23 | 29 | 11.9 | -2.3 | -6.0 | 52% |
| NMAF | SG | 15 | 21 | 1.1 | -4.1 | -0.8 | 52% |
| CMAF | SG | 15 | 29 | 8.4 | 2.3 | -6.7 | 52% |
| SMAF | LW | 7 | 9 | 5.8 | -13.3 | 1.8 | 56% |

**Table 2.** The best performing algorithm parameters for the Thyolo, Muona and Malombe faults based on a misfit analysis using the TanDEM-X DEM. Lowess (LW) or Savitzky-Golay (SG).

# 6 Indicators of structural fault segmentation

## 6.1 Bilila-Mtakataka

In agreement with the findings from Hodge et al. (2018a), the distribution of scarp height - a proxy for the vertical displacement (King et al., 1988; Jackson et al., 1996; Keller et al., 1998; Hetzel et al., 2004) - defines six major (first-order) structural segments along the Bilila-Mtakataka fault (Fig. 10). Scarp slope is less variable than previously considered (Hodge et al., 2018a), especially within the Citsulo segment (Fig. 10c). This is likely due to the lower spatial resolution of measurements used in previous studies, where poor quality measurements - unrepeatable and inaccurate due to the reasons given in Section 2.1 - greatly influenced the along-strike profile. The ability to measure scarp parameters at a high spatial resolution is a major benefit of an automated algorithm. Using the traditional, manual approach, increasing the number of fault scarp profiles would dramatically increase the time required.

In addition, by increasing the spatial resolution of measurements, along-strike changes in displacement may be identified at a smaller scale. As regular, frequent spacing cannot account for scarp height differences caused by local geomorphology (i.e. erosion, deposition, non-fault related landforms), many of the measurements and signals may not be entirely tectonic (Zielke et al., 2015). A moving mean is therefore used to minimise such local influences. In fig. 10 the moving mean window size is set to 1 km for the Pleiades algorithm results. The general trend of the algorithm results still follows the manually derived trend taken using a larger window size, but variations in height occur along-strike at an even smaller scale than previously considered, as detailed below.

Changes in scarp height with a magnitude larger than the typical algorithm error (≥ 5 m) are considered to be real along-fault changes in scarp morphology. As the algorithm assumes only a single scarp surface, multi-scarps (also known as multiple scarps) or composite scarps associated with individual ruptures (Wallace, 1977; Nash, 1984; Crone and Haller, 1991; Zhang et al., 1991; Ganas et al., 2005), will be treated as a single scarp. In other words, the calculated scarp height is the cumulative





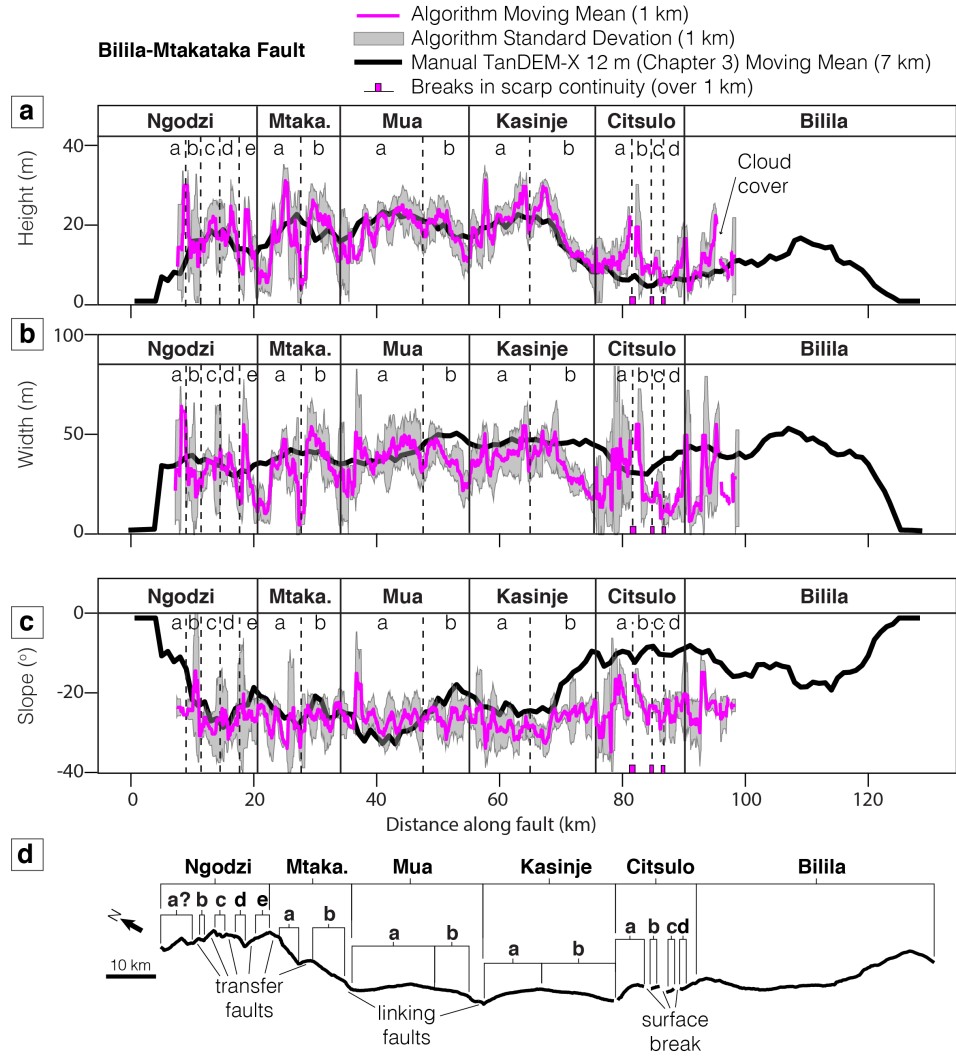

**Figure 10.** Panels a to c) Height, width and slope profiles for the Bilila-Mtakataka fault scarp using the Pleiades DEM, indicating the major segments proposed in Hodge et al. (2018b) (Ngodzi, Mtakataka etc.) and newly identified secondary segments (a, b etc.) from this study. d) A map-view showing fault structural segmentation, breaks in scarp and the location of inferred linkage structures.





vertical displacement at the surface. The results indicate that (second-order) secondary structural segments exist along the Bilila-Mtakataka fault, as typically expected for a large, structurally segmented fault (e.g., Walsh and Watterson, 1990, 1991; Peacock and Sanderson, 1991, 1994; Trudgill and Cartwright, 1994; Dawers and Anders, 1995; Manighetti et al., 2015). Faults forming hard-links between major segments, and those linking secondary segments, are also observed and we discuss specific

examples below.

For the Ngodzi segment, at least five small (2 to 5 km long) secondary segments, joined by high-angled linkage structures, are identifiable by the local highs and lows in scarp height (Fig. 11a). The separation-to-length ratio between each secondary segment is around $\sim 1$, an ideal geometry for a transfer fault to establish (e.g., Bellahsen et al., 2013; Hodge et al., 2018b). The scarp appears to splay at the intersection between the southern most Ngodzi secondary segment and the Mtakataka segment,

potentially comprising a single, or series of, small transfer faults (Fig. 11a). A small rural settlement exists on top of the elevated surface caused by the footwalls of the two major segments; this has lead to a significant amount of erosion to the scarp face making it difficult to identify a hard-link between the major segments (Fig. 11b).

The intersection between two parallel, slightly offset secondary segments on the Mtakataka segment is distinguishable by a low in scarp height. The sharp change in scarp trend at this intersection suggests the existence of a high-angled transfer fault.

The Mtakataka and Mua segments are then linked by a $\sim 2$ km long linking fault angled on average $\sim 35°$ from the scarp trend. The geometry between segments is most favourable for a fault bend (e.g., Jackson and Rotevatn, 2013; Hodge et al., 2018b). Furthermore, there is no evidence of a breached relay ramp.

The height of the fault scarp along the Mua segment is indicative of a single, major structural segment (i.e. bell-shaped height/displacement-length profile with slip maximum at the centre); however, a small decrease in height at $\sim 47$ km may be ev-

idence of a inter-segment zone between secondary structural segments (Fig. 11c). If so, the subtle change in scarp morphology suggests that the secondary segments initiated as separate faults but have since hard-linked and matured, as the displacement deficit is minor. At the southern tip of the Mua segment, there is a decrease in height ($\sim 10$ m) and change in geometry several kilometres from the northern tip of the Golomoti segment, which is marked by the river Livelezi (Fig. 11c). The river itself marks the only break in scarp continuity between the Mua and Kasinje segments. The $> 45°$ change in scarp trend and slight

overlap between segments, suggest that the offset may have been bridged with a relay ramp that has since breached, forming a hard-link and subsequently been exploited by the Livelezi River. Similar to the Mua segment, the displacement distribution along the Kasinje segment is characteristic of a single, major segment, but a local decrease in scarp height ($< 5$ m) at $\sim 63$ km suggests that two secondary segments may have once existed as isolated structures (Fig. 11c). These segments have since hard-linked, matured and the cumulative displacement has reduced much of the deficit within the inter-segment zone.

Previous work has suggested that the Citsulo segment had a general zone of scarp discontinuity stretching $\sim 10$ km in length (Hodge et al., 2018a). Here, we find evidence of several small breaks along the fault scarp within the Citsulo segment (Fig. 11d). Breaks are up to 2 km in length suggesting that the Citsulo segment comprises several small ($\sim 2$ km), en echelon secondary segments.







**Figure 11.** Oblique perspective images taken from the TanDEM-X and Pleiades DEMs for the Bilila-Mtakataka fault. a) Ngodzi segment normal (nf) and transfer faults (tf) trend in a zig-zag pattern. b) Mtakataka segment normal and transfer faults. c) Mua and Kasinje segments intersecting at the river Livelezi. A small increase in scarp height on the Mua segment may relate to a relay ramp linkage. d) The Citsulo segment and area of discontinuity. Small, north-striking, left-stepping faults are offset by up to 1 km. Example profiles for SRTM, TanDEM-X and Pleiades DEMs are also shown.





### 6.1.1 Thyolo and Muona

Fig. 12 shows the along-strike profile for the Thyolo and Muona faults. Scarp slope for both Thyolo and Muona faults is fairly uniform, averaging around $\sim 22°$ with a small standard deviation $< 5°$ (Fig. 12c). Scarp height and width, however, show more variation along-strike (Fig. 12a,b). We interpret three major segments along the TOF from the numerous peaks and troughs in

scarp height, called TOFS1, TOFS2 and TOFS3, whose lengths are between 15 km and 30 km. In contrast, the height of the shorter MOF is fairly consistent before it tapers off toward the southeastern fault end. We therefore interpret the Muona fault to consist of a single major segment. Below we describe each major segment and any associated secondary segments and linkage structures. The faults do not appear hard-linked, likely due to the large separation-to-length ratio ($\gtrsim 0.1$), which may favour continued along-strike growth or a transfer-style link (e.g., Bellahsen et al., 2013; Hodge et al., 2018b). Below we describe

each major segment of the faults and any associated secondary segments and linkage structures.

For both TOFS1 and TOFS2, the distribution of scarp height is bell-shaped with slightly asymmetry of the TOFS2 profile toward the inter-segment zone. For TOFS1, scarp height is larger and increases from $\sim 10$ m at the segment ends to $\sim 30$ m at the centre; an increase in width is also observed at the centre, resulting from the consistent scarp slope. The maximum height of the TOFS2 scarp is $\sim 20$ m. For both, the peaks in scarp height coincide with the apex of the convex geometry of the fault

scarp (Fig. 12d). The scarp height and width of TOFS3 increases gradually toward the southeast, where the segment extends into the footwall of the MOF. The scarp height of TOFS3 within the overlapping zone between the Thyolo and Muona faults exceeds the MOF scarp height by, on average, $\sim 5$ m. The standard deviation of measurements here is larger than elsewhere along both fault scarps, indicating intense local variability in scarp parameters.

The low count of scarps recognised by the algorithm along the Thyolo fault meant that we cannot conclusively interpret

the existence of secondary segments. There are several $> 1$ km long breaks in where the algorithm could recognise a scarp along TOFS1 and TOFS2; however, the distribution of scarp heights does not conclusively imply second-order segmentation. For TOFS3, several major breaks in scarp continuity coincide with sharp changes in scarp trend. Based on these changes in trend, we interpret three secondary segments, called TOFS3a, TOFS3b and TOFS3c, and associated linkage structures (Fig. 14a). Each of these secondary segments has a length $\sim 10$ km and TOFS3c coincides with the length of the overlapping zone

between Thyolo faults. There is no conclusive evidence of secondary segments along the Muona fault. Two major $\sim 4$ km breaks in scarp continuity toward the segment end suggest a shorter fault scarp ($\sim 20$ km) than our manual analysis suggested. Large gaps between profiles, typical of a manual analysis, may therefore fail to account for small-scale changes in morphology and over/under-estimate fault lengths.

### 6.1.2 Malombe

In agreement with the manual analysis, the slope of the NMAF and CMAF fault scarps are remarkably similar, averaging $\sim 18°$ (Fig. 13). Based on the remarkably uniform scarp height, averaging $\sim 8$ m, the NMAF appears to comprise a single major segment. A small break in scarp continuity and $\sim 10$ m decrease in scarp height along the CMAF at $\sim 24$ km suggest an inter-segment zone between two major segments, called CMAFS1 and CMAFS2 (Fig. 14b). The scarp height of CMAFS1



**Figure 12.** Panels a to c) Height, width and slope profiles for the Thyolo and Muona fault scarps using the TanDEM-X DEM. d) A map-view showing fault structural segmentation, breaks in scarp and the location of inferred linkage structures.




is the largest of all Malombe faults, averaging $\sim 8$ m. The distribution of scarp height along the CMAFS1 is roughly bell-shaped with an asymmetry leaning toward the NMAF. The height of the short CMAFS2 segment decreases by around 1 m per kilometre from north to south. A major $\sim 6$ km break in the SMAF scarp continuity implies either two major segments, SMAFS1 and SMAFS2, or a continuous deeper fault that has not broken the surface continuously. The scarp height for the

SMAF is relatively constant, averaging $\sim 5$ m, and does not display a bell-shaped profile. No secondary segments were inferred from the distribution of scarp height along any Malombe fault scarp. The longest segment, CMAFS1 (18 km), does comprise several breaks in scarp continuity and changes in morphology typical of second-order segmentation, however, a higher spatial resolution of measurements would need to confirm this.

## 7   Discussion

### 7.1   Algorithm performance

In this study we developed an algorithm for calculating the height, width and slope of a fault scarp from scarp elevation profiles: Scarp PARameTer Algorithm (SPARTA). A series of sensitivity analyses were performed using a synthetic catalogue prior to using the algorithm on real fault scarps. The benefits of creating a synthetic catalogue are two-fold: (1) a vast number of scarp profiles can be built to improve the performance of the algorithm through an in-depth misfit analysis; and (2) by creating a

synthetic catalogue that mimics the typical fault scarp morphology of interest, and performing a sensitivity test for resolution, the benefits of high resolution satellite data can be assessed prior to purchasing costly data (see the supporting information for synthetic catalogue test results). The synthetic catalogue should mimic the typical fault scarp morphology of interest. This can be achieved by selecting *a prior* catalogue parameters based on initial findings using a free, low resolution data DEM (e.g., SRTM). The general morphology of the fault scarp and climatic conditions heavily influence the chosen catalogue parameters.

For example, for regions where transport-limited fault scarps and vegetation are typical, the catalogue parameters can include diffusion and noise. In contrast, for regions typical of diffusion-limited fault scarps and limited vegetation, no diffusion and less noise can be used.

We found that the major influence on algorithm performance was the signal-to-noise ratio within the elevation profiles. Profiles containing a low signal-to-noise ratio will likely require the inclusion of a filter within the algorithm. In contrast, the

algorithm may perform well without a filter for profiles with a high signal-to-noise ratio. In general, the algorithm was able to calculate scarp height and slope with a smaller misfit, compared to a manual analysis, than scarp width. The performance was improved by calculating scarp width based on the estimated scarp height and slope, rather than directly (Fig. 9b). However, this approach assume scarp planarity and therefore precludes use of the results for scarp degradation analysis or interpretation of single-rupture versus composite scarps.

In our case studies, the percentage of fault scarps where the algorithm was able to identify the scarp varied between $\sim 50\%$ and $\sim 80\%$. Lower returns coincided with fault scarps identified manually to contain large breaks in scarp continuity. Although the algorithm selects the best performing parameters from the misfit analysis, individual profiles may still fit poorly. Quality





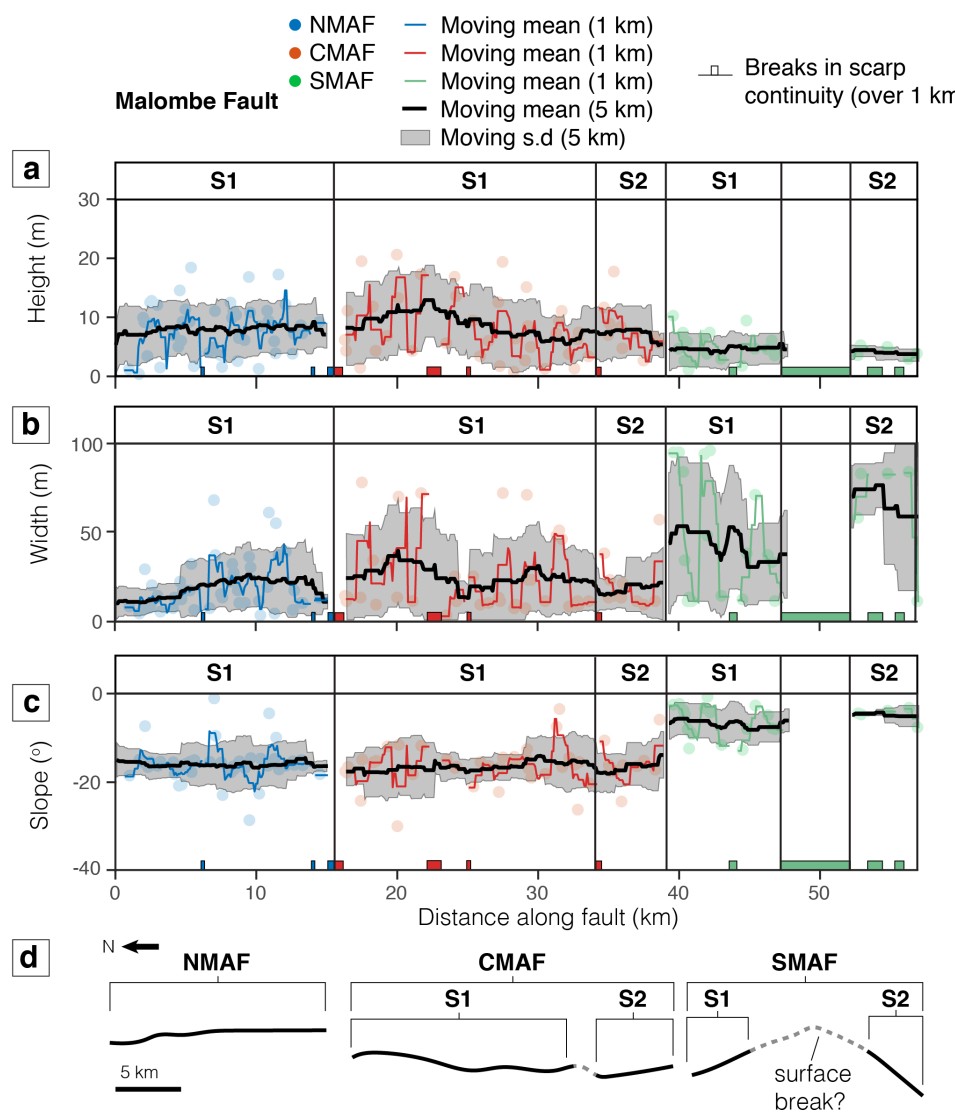

**Figure 13.** Panels a to c) Height, width and slope profiles for the Malombe fault scarps using the TanDEM-X DEM. d) A map-view showing fault structural segmentation, breaks in scarp and the location of inferred linkage structures.





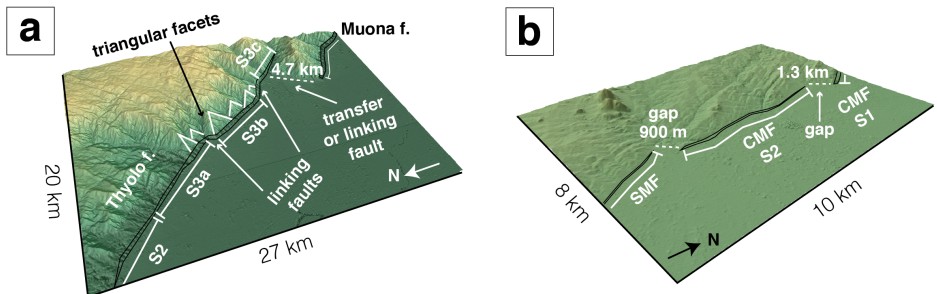

**Figure 14.** Oblique perspective images taken from the TanDEM-X DEM for sections of the a) Thyolo and Muona faults and b) Malombe fault scarps. a) The secondary segments along TOFS3 of the Thyolo fault, showing the triangular facets synonymous with an mature scarp, and the structure that connects the Thyolo and Muona faults. b) The soft-linkage between the central fault segments (CMAFS1 and CMAFS2) and between the central and southern faults, each are offset by around 1 km.

checks were applied to remove outliers and improve the results, but this decreased the number of identified scarps for each case study to between $\sim 15\%$ and $\sim 55\%$ of profiles.

The performance of the algorithm was not significantly affected by DEM resolution, but a number of differences were apparent between datasets (see the supporting information for a more detailed discussion). The lower the DEM resolution,

the smaller the number of identifiable fault scarps, but the smaller the standard deviation of parameters. We found that a 30 m resolution DEM identified on average 20% fewer fault scarps than a high resolution 5 m DEM. Scarp width and slope calculated by the algorithm were on average wider and more gentle using a low resolution DEM. In general though, we found that for these southern Malawi faults, the use of expensive, high resolution DEMs in quantifying large-scale changes in scarp height over the scale of an entire fault, did not bring any additional benefits over using a medium- or low-resolution DEM. An

exception is where scarp height is smaller than the elevation changes produced by background noise such as vegetation. This is an important finding if using this algorithm to study fresh ruptures, which are apparent as steep faces of fault scarps (Wallace, 1977), or scarps whose vertical displacement is less than 10 m, for which we recommend using a very high ($\leq 1$ m) resolution DEM and a large slope $\theta$ threshold.

Although our algorithm performed well against a number of manual analyses, the algorithm has some limitations including

the reliance on manually picking the fault scarp trace. As low resolution DEMs smooth small-scale changes in scarp trend, this is most pertinent when using a high resolution DEM and a high spatial frequency of sample points (Fig. 6). In addition, we have here used scarp-perpendicular scarp profiles, which may not be appropriate for oblique slip faults or sections of the scarp that trend at a high-angle to the slip vector (Mackenzie and Elliott, 2017). Slip vectors could not be measured for the southern Malawi faults (Hodge et al., 2018a). Using the regional extension direction, the total surface slip may not be truly represented

by the scarp height for the northern BMF segments, nor the Thyolo and Muona faults. If the slip vector of a fault is known, this can be accounted for in the algorithm.




We found that the distance between nodes (vertices of the scarp trace) should not exceed an order of magnitude above the horizontal resolution of the DEM. However, as long as the large-scale fault trend is correctly chosen, a wide profile length $x$ (here set to around four times the largest scarp width) should cover a sufficient amount of the upper and lower original surfaces for the algorithm to calculate the scarp height correctly. In addition, as the algorithm uses a fixed slope threshold, if
there is a lot of noise within the data, or there is a significant amount of heterogeneity in the scarp's morphology along-strike, small, or gently dipping, fault scarps may not be identified by the algorithm. This can be alleviated by either: (1) identifying morphologically different parts of a fault scarp and running the algorithm on these profiles separately, as we have done for Malombe; or (2) following the first algorithm run, running the algorithm again on poorly resolved regions, including a manual analysis to identify the best algorithm parameters to use. We suggest that the algorithm may face additional limitations in a
more complex or varying terrain than considered here.

### 7.2    Normal faults in southern Malawi

As fault scarps are indicative of past earthquake events (Wallace, 1977), we use our geomorphological findings to better understand the rupture history for each fault. Assuming that a scarp is formed from a single earthquake event, the average scarp height can be used as a proxy for average coseismic slip (e.g., Morewood and Roberts, 2001) to calculate the slip-length
ratio (Scholz, 2002). The typical global slip-length ratio range for a single earthquake is $10^{-5}$ to $10^{-4}$ (Scholz, 2002). Note, however, that fault slip at the surface may be several times less than the slip at depth (e.g., Villamor and Berryman, 2001). We simplify the length value to be the straight-line distance between the tips of the surface trace, which is less than the length of the irregular surface trace. Here, we found that the $\sim 65$ km long Thyolo and $\sim 110$ km long Bilila-Mtakataka faults have scarp heights that average $\sim 20\pm11$ m and $\sim 17\pm7$ m, respectively. The average scarp height of the Bilila-Mtakataka fault
found here is larger than found previously (e.g., Jackson and Blenkinsop, 1997; Hodge et al., 2018a), but this is because only $\sim 90$ km of the fault was analysed and the non-analysed sections of the Bilila-Mtakataka fault, predominantly the $\sim 35$ km long Bilila segment, have a smaller scarp height. Due to the close agreement between algorithm and manual calculations, we hereafter use the average scarp height from (Hodge et al., 2018a) ($11\pm7$ m). The average scarp height of the $\sim 28$ km long Muona fault and the $\sim 55$ km long Malombe fault was found to be $\sim 10\pm5$ m and $\sim 7\pm5$ m, respectively. If each scarp is
representative of a single earthquake event, then the average slip-length ratios for each fault (1 - $4\times10^{-4}$) fall on or above the upper limit of the typical global range (Table 3; Scholz, 2002). To account for errors in fault length measurements we apply an uncertainty of 2 km.

     Whilst large slip-length ratio values are rare (Middleton et al., 2016), they have been calculated for the 1897 $\sim$ M$_W$ 8.1 Assam earthquake ($2.2\times10^{-4}$; Bilham and England, 2001), the 2001 $\sim$ M$_W$ 7.6 Bhuj earthquake ($3\times10^{-4}$; Copley et al.,
2011) and the M$_W$ 7.6 1999 Chi-Chi earthquake ($1\times10^{-4}$; Lee et al., 2003); however, none of these earthquakes occurred on normal faults and all exist within regions of higher strain rate than the EARS. In comparison, the only well-documented event we could find with a recorded slip-length ratio within the EARS was for the $\sim$ Ms 6.8 1928 Kenya earthquake (Ambraseys and Adams, 1991), whose 1 m scarp could be traced for $\sim 38$ km at the surface (Ambraseys, 1991b), resulting in a ratio of $\sim 2.8\times10^{-5}$.



**Table 3.** Slip-length ratios for southern Malawi faults: Bilila-Mtakataka (BMF), Thyolo (TOF), Muona (MOF) and Malombe (MAF)

| Fault Name | Fault Length $L$ (km) | First-order Segment Lengths (km) | Average Scarp Height $\bar{H}$ (m) | Average slip-length ratio ($\times 10^{-5}$) | Slip-length ratio range ($\times 10^{-5}$) |
|---|---|---|---|---|---|
| BMF | 110 | 13 - 38 | 17±7 | 16 | 9 - 22 |
| TOF | 65 | 18 - 27 | 20±11 | 31 | 13 - 49 |
| MOF | 28 | 28 | 10±5 | 36 | 17 - 58 |
| MAF | 55 | 5 - 18 | 7±5 | 13 | 4 - 23 |

Abnormally large slip-length ratios may be a result of overestimating surface slip, as shown by Middleton et al. (2016) for the $\sim M_W$ 7.3 1739 Yinchuan earthquake in China, whose original slip-length estimate was $1.3 \times 10^{-4}$. They recalculated this value to be $3.8 \times 10^{-5}$ based on a slightly shorter surface rupture length (87 km compared to 88 km) and a smaller average slip value (3.3 m compared to $\sim$ 12 m). Thus, the new slip-length ratio is within the global range (Scholz, 2002). Here, even

when accounting for measurement errors within the satellite data and algorithm calculations, we find that each of our southern Malawi fault scarps have slip-length ratios larger than the global mean (Table 3; Scholz, 2002).

### 7.2.1    Number of events

The slip-length ratio calculation uses the assumption that the current scarp was formed by a single earthquake event. Therefore, the large values for our southern Malawi faults either are a result of local effects, such as large seismogenic thickness (Jackson

and Blenkinsop, 1993), or suggest that each scarp has been produced by multiple earthquake events. Whether the current scarps were each formed by single, large slip rupture, or multiple, smaller slip ruptures is an important question for assessing the seismic hazard in the region. As the surface length is well constrained, and are in fact smaller than the longest faults in the EARS (e.g., Vittori et al., 1997; Morley, 1999), the validity of the slip-length ratios are governed by the scarp height for each fault (Table 3).

Well-documented, historically-recorded continental normal fault scarps formed by single earthquake events typically have a height less than 10 m (Zhang et al., 1986; Walker et al., 2015). A short, incomplete earthquake catalogue (Midzi et al., 1999), and slow extension rates along the Malawi Rift (Stamps et al., 2008; Saria et al., 2014) leading to long recurrence intervals (Hodge et al., 2015) mean that there is a lack of recorded earthquake events in the Malawi Rift with visible surface offsets. Historical earthquakes that have occurred in the Malawi Rift, either did not rupture the surface, such as the 1989 $\sim M_W$ 6.1



Salima earthquake (Jackson and Blenkinsop, 1993), or small ($<$ 1 m) amounts of surface displacement, such as the 2009 $M_s$ 6.2 Karonga sequence (Biggs et al., 2010; Macheyeki et al., 2015). The latter resulted in an average scarp height of $\sim$ 10 cm and surface rupture length of 9 km. There are a number of reported events however within the EARS, but outside the Malawi Rift, that have been suggested to have produced significant ($>$ 10 m) vertical displacement. For example, within the Rukwa

Rift, just north of the Malawi Rift, there is evidence of a Late Pleistocene earthquake producing $\sim$ 10 m of uplift in the Songwe valley, Rukwa (Hilbert-Wolf and Roberts, 2015). Constraining this displacement to a single event however is challenging due to its age. This event occurred within the same region reported to have hosted one of the largest recorded earthquakes on the EARS, the 1910 $\sim$ M 7.4 Rukwa earthquake (Ambraseys, 1991a). The most likely fault to have hosted this event is the Kanda fault, which has a reported maximum scarp height of 50 m (Vittori et al., 1997). The Kanda scarp is reported to comprise a

fresh face synonymous with a recent rupture (Vittori et al., 1997); but due to the a lack of absolute age estimates on the Kanda fault scarp, and because the region has experienced frequent earthquakes since the Late Pleistocene (Hilbert-Wolf and Roberts, 2015), its unclear whether this scarp was formed by a single event. More modest scarp heights such as the 1.5 m scarp along the $\sim$ 50 km Katavi fault have been recorded in the Rukwa Rift (Kervyn et al., 2006). The Katavi fault however is considered to be a possible aftershock site resulting from the 1910 event (Kervyn et al., 2006) and does not reflect a main earthquake event.

Using the global mean slip-length ratio of $5\times10^{-5}$ (Scholz, 2002), and assuming slip on each fault is pure normal, the number of events required to generate the current scarp heights along the Bilila-Mtakataka, Thyolo, Muona and Malombe faults is between 2 and 5, with the Thyolo fault requiring the greatest number of events. This does not account for vertical erosion between events and therefore may be an underestimate.

### 7.2.2 Displacement profile and segmentation

Fault scarps developed through multiple events have been observed in many regions (Wallace, 1977; Nash, 1984; Crone and Haller, 1991; Zhang et al., 1991; Ganas et al., 2005). Multiple earthquake events have also been suggested as a method for fault development, where large faults form iteratively through fault growth and linkage of smaller, fault segments (e.g., Peacock and Sanderson, 1991; Cowie and Scholz, 1992; Anders and Schlische, 1994).

The along-strike pattern of scarp height for the Bilila-Mtakataka (at least up to the Citsulo segment) and Malombe fault

scarps show a symmetrical bell-shaped profile, with the maximum scarp height near the centre of the fault (e.g Walsh and Watterson, 1987, 1990; Peacock and Sanderson, 1991; Manighetti et al., 2001; Nicol et al., 2010), whereas the Thyolo fault displays a distinctive asymmetric, triangular appearance (e.g., Manighetti et al., 2001, 2009, 2015; Nicol et al., 2005; Soliva and Benedicto, 2004). Height along the Thyolo fault scarp decreases southeastward before increasing toward the overlap zone with the Muona fault. Geological maps indicate that there may be a physical connection between the Thyolo and Muona faults

(Habgood et al., 1973). The triangular distribution and tapering of scarp height along the Thyolo fault scarp may denote that the direction of long-term fault propagation is southeastward onto the Muona fault (e.g., Manighetti et al., 2001, 2015).

By observing the along-strike variation in scarp height for each fault, we found evidence for structural segmentation on each fault. We found that the $\sim$ 110 km long Bilila-Mtakataka fault comprises six major segments, the $\sim$ 70 km long Thyolo fault three, and the $\sim$ 25 km long central Malombe fault two. The Muona fault did not show signs of along-strike segmentation and





is considered a single major segment. Segments along the Thyolo fault and Bilila-Mtakataka fault, with the exception of fault splays within the Citsulo segment, have hard-linked. These hard-links imply fault maturity (Trudgill and Cartwright, 1994; Young et al., 2001). In contrast, gaps between the three Malombe fault segments indicate soft-linkage (Walsh and Watterson, 1991). Our results are consistent with findings from other parts of the EARS, which suggest that the major faults are segmented

at least to the first-order (Ambraseys and Adams, 1991; Manighetti et al., 2015). For example, the ~ 180 km long Kanda fault comprises at least three major, hard-linked segments (Ambraseys and Adams, 1991).

In addition, the increase in spatial resolution in this study, a benefit of an automated approach, meant that secondary segments and linking structures could also been identified for the Bilila-Mtakataka and Thyolo faults. Each major segment along the Bilila-Mtakataka fault scarp comprised between two and five secondary segments, whereas (three) secondary segments were

only identified on the southern-most major segment of the Thyolo fault. Thus, the number of secondary segments, where identified, is consistent with the number found on normal faults in Afar, further north in the EARS (Manighetti et al., 2015). We also found that the length of the major segments correlated with the length of the fault (Table 3). If we consider that these faults grow by linkage of smaller structures (e.g., Peacock and Sanderson, 1991; Cowie and Scholz, 1992; Anders and Schlische, 1994), the existence of fault segments along each fault is evidence of multiple earthquake cycles.

The accumulation of displacement at the segment tips and/or hard-links suggests that each fault has hosted ruptures that have propagated across adjacent segments (e.g., Peacock and Sanderson, 1991; Cartwright et al., 1995). Multi-segment ruptures have been attributed to some of the largest earthquakes on the continents; for example, the ~ $M_W$ 8 1889 Chilik earthquake (Abdrakhmatov et al., 2016). For normal faults, rupture propagation may continue across gaps as large as 10 km (e.g., Biasi and Wesnousky, 2016). The Malombe fault is the only fault studied here with persistent gaps along its surface trace; however,

these gaps are less than 10 km, and may be controlled by the changes in lithology. Some of the gaps coincide with calc-silicate granulite outcrops, which were also observed to cause discontinuities along the BMF (Hodge et al., 2018a). Discontinuous scarps are also a common occurrence of many earthquakes; for example, the ~ Ms 6.9 1928 Laikipia-Marmanet earthquake resulted in a discontinuous surface rupture (Ambraseys and Adams, 1991). No gaps in scarp continuity greater than 5 km were found on either of the Thyolo or Muona faults, and even the Citsulo segment on the Bilila-Mtakataka fault comprises small en

echelon scarps separated by distances of less than 5 km.

### 7.2.3 Scarp age

The exact number and age of historical ruptures on each fault is unknown and requires dating and/or trenching to give an accurate estimation. By using the scarp geomorphology however, a relative age between each scarp can be estimated (e.g., Nash, 1984; Stewart and Hancock, 1990; Avouac, 1993).

The calculated slopes for each scarp lack the steep values that are compatible with a fresh scarp face, typically > 30° for high resolution DEMs (Middleton et al., 2016). This may imply that sufficient time has passed for the scarps to have undergone extensive degradation (e.g., Andrews and Hanks, 1985; Avouac, 1993; Carretier et al., 2002; Tucker et al., 2011). However, as scarps with slopes greater than 30° were found on the Bilila-Mtakataka fault using the Pleaides 50 cm DEM (Fig. 2c), the gentle values are more likely a result of the profile resolution. Slope measurements are therefore fundamentally dependent on





sampling or DEM resolution. Our resolution analysis on the Bilila-Mtakataka fault confirms this, a higher resolution DEM led to steeper average slopes being calculated, compared to a lower resolution DEM (see supporting information). This effect of resolution means that the exact slope values could not be calculated for Thyolo, Muona or Malombe. Despite this, if the diffusivity constant $\kappa$ is similar for each fault scarp - a reasonable assumption given the similar lithology and climatic effects

on each scarp - then the relative differences in the average slope between scarps found using the TanDEM-X DEM may be used to find the relative age difference (e.g., Nash, 1984; Avouac, 1993).

Our TanDEM-X results show that the slope of the Bilila-Mtakataka, Thyolo and Muona fault scarps are similar ($\sim 20°$ on average) and relatively constant along-strike (standard deviation is $10°$), whereas the slope of the Malombe fault scarp is more gentle ($< 20°$) and becomes even more gentle toward its southern end. The gentle slope of Malombe scarp therefore

may suggest that its most recent rupture (that broke the surface) may be prior to recent events on the Bilila-Mtakataka and Thyolo faults. However, this may also be due to the lithology of the Malombe fault scarp, which for most of its northern end is felsic paragneiss. In contrast, the lithology of the Bilila-Mtakataka, Muona and Thyolo fault scarps is predominantly mafic paragneiss, with local variability on the Bilila-Mtakataka (e.g., Habgood et al., 1973; Hodge et al., 2018a). The gentler slope on Malombe may also be related to the TanDEM-X resolution, which resolves the scarp over several fewer pixels than for

Bilila-Mtakataka, Thyolo and Muona.

Each fault segment along the Thyolo fault has hard-linked, indicating that the fault is mature. In contrast, the Malombe fault and the Citsulo segment on the Bilila-Mtakataka fault both comprise several gaps in scarp continuity and soft-linkages synonymous with a less mature fault (Walsh and Watterson, 1991). The structural evidence, and its position between two opposite-dipping border faults (BMF and Mwanjage), suggests that the Malombe fault is an intra-basin fault whose develop-

ment is closely related to the Bilila-Mtakataka's, whilst the Thyolo and Muona faults are older, more mature structures.

### 7.2.4   Earthquake magnitude

The Bilila-Mtakataka fault has the longest scarp in this study, with a total surface trace measuring $\sim 110$ km in length. The second longest scarp trace is this study was the Thyolo fault, which measured $\sim 70$ km in length. The length of the Muona fault was $\sim 25$ km. The length of each Malombe fault scarp is between 15 and 25 km, with a total cumulative length of $\sim 50$

km. Whereas the more mature northern part of the East African Rift System (EARS) comprises faults whose maximum length is $\sim 65$ km and median length is 10 km (Manighetti et al., 2015), the Bilila-Mtakataka and Thyolo faults are more comparable to the large fault scarps observed on the western and eastern branches of the EARS, such as the 140 km long Lokichar fault in the Kenya Rift (Morley, 1999) and the 180 km long Kanda fault in the Rukwa Rift (Vittori et al., 1997). In addition, the thick ($\sim 40$ km) seismogenic layer in southern Malawi (Jackson and Blenkinsop, 1993) implies that the down-dip fault width is also

large (Wallace, 1989).

Of primary concern is the seismic hazard posed by these faults, as empirical relationships (e.g., Wells and Coppersmith, 1994; Hanks and Kanamori, 1979) suggest that the larger the fault, the larger the maximum earthquake magnitude. It has been suggested that the most recent earthquake on the Bilila-Mtakataka fault ruptured its entire length, an event that would equate to a $\sim M_W$ 8 earthquake (Jackson and Blenkinsop, 1997). Using the equation $M_W = \frac{2}{3} \cdot \log(G\alpha L^2 W) - 6.05$ (Aki, 1966;





Hanks and Kanamori, 1979), where $G$ is the modulus of rigidity (here taken as 30±5 GPa, e.g., Crider and Pollard, 1998; Biggs et al., 2009), $\alpha$ is the slip-length ratio (see Table 3), $L$ is the fault length (Table 3), $W$ is the fault width, and the fault dip is $\delta = 60°$ - the moment magnitude $M_W$ for each fault can be found. We assume here that the rupture occurs through the full thickness of the seismogenic zone, and as such is calculated using $W = Z_{ST}/\delta$, where the seismogenic thickness $Z_{ST}$ is

40±15 km (e.g., Jackson and Blenkinsop, 1993). By accounting for uncertainties within the parameters a $M_W$ range is given. A complete rupture of the Bilila-Mtakataka, Thyolo, Muona and Malombe faults would equate to a $M_W$ range of 7.9 - 8.4, 7.7 - 8.3, 7.2 - 7.9 and 7.2 - 8.0, respectively. Assuming the average subsurface displacement is 1.6 times greater than the average surface displacement (Villamor and Berryman, 2001), the maximum $M_W$ increases to 8.5, 8.4, 8.0 and 8.1 in the respective order above.

Whilst large magnitude strike-slip and reverse-slip subduction zone earthquakes have been known to produce surface ruptures with lengths comparable to these southern Malawi scarps (e.g., $M_W$ 8.1 1855 Wairarapa earthquake; Rodgers and Little, 2006, and $M_W$ 8.1 2001 Central Kunlun earthquake; Lin, 2002), observations of continental normal or reverse earthquakes producing such surface rupture lengths are rare. Examples include the $\sim$ M 8 1556 Huaxian (Yuan et al., 1991) and $\sim$ M 8 1739 Yinchuan events (Zhang et al., 1986; Deng and Liao, 1996), both in central China, and the $\sim$ $M_W$ 7.7 Egiin Davaa

earthquake in central Mongolia (Walker et al., 2015). The only EARS event that may have resulted in a surface rupture with length of similar magnitude to our fault scarps is the 1910 $\sim$ M 7.4 earthquake in the Rukwa region of Tanzania (Ambraseys, 1991a), which had a magnitude similar to our estimates above.

Not all large magnitude earthquakes produce a surface rupture, and not all earthquakes rupture the entire fault length. Many of the largest recorded earthquakes along the EARS, including the 1990 $\sim$ Ms 7.2 southern Sudan earthquake (Ambraseys and

Adams, 1991) and the $\sim$ $M_W$ 6.8 2005 Lake Tanganyika earthquake (Manyele and Mwambela, 2014), lack a corresponding scarp. Even the subsurface rupture lengths of these events have been modelled to be just $\sim$ 26 km and $\sim$ 16 km, respectively (Moussa, 2008), significantly smaller than the total lengths of each of fault scarps in this study. In addition, one of the few recorded surface ruptures for a large magnitude event along the EARS, the $\sim$ Ms 6.9 1928 earthquake on the Laikipia-Marmanet fault in Kenya - the largest instrumentally recorded earthquake in the Kenya rift - resulted in just a $\sim$ 38 km long surface rupture

(Ambraseys, 1991b).

As all faults but the Muona fault comprise several structural segments, ruptures that terminate at the geometrical ends of each structural segment (i.e. a single-segment rupture), or ruptures that occur across multiple segments but not the whole fault (i.e. multi-segment rupture), may occur on each fault. The geomorphology on each also shows evidence for segmented ruptures. The triangular slip distribution on the Thyolo fault may be evidence of segmented ruptures (Manighetti et al., 2001, 2005),

the discontinuity at the Citsulo segment on the Bilila-Mtakataka fault may be evidence that the fault is actually two discrete structures, and the soft-linked Malombe fault segments may also rupture individually. Using the moment magnitude equations and the average scarp height for each structural segment, single-segment ruptures (with lengths between 20 and 40 km) on each of fault would generate an earthquake with a $M_W$ between 6.8 $\sim$ 8.1 if the earthquake ruptures the entire down dip width, or 6.7 $\sim$ 8.0 if the rupture width is constrained to be less than the rupture length (i.e. 20 km). Therefore, single-segment ruptures

on each fault can still generate earthquakes with magnitudes comparable to the largest events recorded within the EARS, and





larger than any historically-recorded earthquake in Malawi. However, as single-segment ruptures with a slip value equal to the average scarp heights measured here imply an even larger slip-length ratio (up to $10^{-3}$), and are therefore unlikely, the question that still needs addressing is whether the scarp of each fault comprises a single rupture, or multiple ruptures.

## 8  Conclusions

In this study, we have developed a semi-automated algorithm for quantifying along-strike variations in scarp morphology: Scarp PARameTer Algorithm (SPARTA). We show that the algorithm performs comparatively well against traditional, manual analyses, but allows for a greater spatial resolution of measurements, improving the understanding of the morphological parameters along a fault scarp. We have shown that DEM resolution does not greatly influence the algorithm's performance when used to infer first-order fault structural segmentation and associated linkage structures. However, a high resolution DEM may

be required to conclusively infer second-order structural segmentation, especially along faults with small scarp heights. For the southern Malawi faults, the distribution of scarp height along-strike, found using our algorithm, indicates that all three of the four faults, Bilila-Mtakataka, Thyolo, and Malombe, comprise first-order segmentation at their surface. The Muona fault is a single, major segment. Using a Pleiades DEM, second-order segmentation is clearly apparent along the Bilila-Mtakataka fault. Assuming the average scarp height reflects the average slip at the surface, if each scarp was formed by a single earthquake

event, the slip-length ratio for each fault exceeds the global upper limit proposed by Scholz (2002). The distribution of scarp height close to, and within, the inter-segment zones for each fault suggests that the Bilila-Mtakataka and Thyolo fault segments have hard-linked, incrementally through several earthquake cycles, and the Malombe faults are soft-linked. Our results suggest that each fault has likely formed through multiple events; however, to constrain the co-seismic slip and rupture length of each event, a detailed study is required for each fault scarp.

*Acknowledgements.*  The data used are listed in the references, tables, and supplementary material. MH is supported by the NERC GW4+ Doctoral Training Partnership (grant code NE/L002434/1) and Centre for Observation and Modelling of Earthquakes, Volcanoes and Tectonics (COMET). JB is supported by COMET, the NERC Large Grant Looking into Continents from Space (LiCS, NE/K010913/1) and the EPSRC Global Challenges PREPARE (EP/P028233/1). AF is supported by COMET and EPSRC PREPARE. AE is supported by COMET and NERC LiCS (NE/K011006/1). HM and FM acknowledge the Geological Survey Department, Malawi for attaching them to the project.

All authors acknowledge the Geological Survey Department, Malawi for their assistance with fieldwork in Malawi. Pleiades data was obtained using a small grant from COMET. TanDEM-X data was obtained via DLR proposal DEM_GEOL0686.

*Code availability.*  The SPARTA algorithm codes are available on GitHub at https://github.com/mshodge/FaultScarpAlgorithm alongside a number of synthetic catalogs.

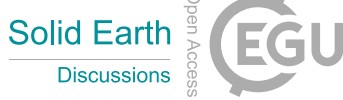



## Appendix A

### A1    Outlier identification

To improve the accuracy of the results obtained using the algorithm, we conduct a number of quality checks. First, algorithm results with negative scarp heights and positive slopes are removed. Next, because misfit values for scarp width were larger

than for scarp height and slope, and scarp width is the primary influence on height and slope calculations, algorithm results where scarp width was twice as large as the maximum found in the manual analysis are also discarded. This value is arbitrary, however, we choose a value above the manual maximum (Fig. 7b) as we do not want to discard wide fault scarps that are real and did not appear in the manual analysis by random chance. Here, this removes all results where the scarp width was greater than $\sim 100$ m. Then, as the algorithm results are approximately normally distributed (black, Fig. 9a), outliers are removed by

applying a threshold, set to $2\sigma$ ($\sim 95\%$ confidence interval) of the remaining data. For the Bilila-Mtakataka fault, these quality checks removed 223 (31%) results and significantly reduced the standard deviation of the remaining data (pink, Fig. 9a). The estimates of average scarp height decreased by 3 m, width dramatically by 47 m, and slope increased in steepness by $3°$.

### A2    Improving width estimate

The results from this natural study corroborate those found in the performance test for the algorithm, and suggest that the

algorithm calculates scarp height with less error and scarp width (Fig. 2). Scarp width can also be calculated as a function of the scarp height and slope, using the equation $W = H/\tan\alpha$. We compare scarp widths and find that they correlate well ($R^2$ = 0.75) for widths of 100 m, or less (Fig. 9b), but scarp widths obtained directly from the algorithm may be an overestimation by up to $\sim 15$ m for widths under 100 m. This may explain why scarp width misfit values were larger than height or width misfit values (Fig. 8). Since no fault scarp on the Bilila-Mtakataka fault was measured to be wider than 100 m, as reported

in Hodge et al. (2018a), nor in the manual analysis in this research, results wider than this may be a result of poor algorithm performance, likely due to a low signal-to-noise ratio. However, as it is difficult to consistently apply an exact angle threshold when manually picking, we don't necessarily expect automated and manual results to be exactly the same. As a result, some differences been manual and automated approaches may be due to the misidentification of scarp crest and base in the manual approach. From hereafter, we calculate scarp width as a function of height and slope. We find that this approach is appropriate

here as we are simplifying the scarp to be planar, but would not be appropriate if adapting this algorithm to calculate other morphological parameters such as scarp/diffusion age.



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
