# Peer review of "A Semi-Automated Algorithm to Quantify Scarp Morphology (SPARTA): Application to Normal Faults in Southern Malawi"

_Solid Earth, 2018_

## Referee Comment (RC1) · Anonymous Referee #1 · 25 May 2018

This paper is an interesting one, which aims to produce an automated method of analyzing DEMs in order to extract metrics about fault scarps. The aim is a good one, in that it will allow scarp geometries to be analyzed with minimal interpretive input, and rapidly over large areas, which will be useful in terms of documenting and interpreting any along-strike variations that are present (e.g. due to segmentation). I have some comments that I hope will help to improve the clarity of the manuscript.

1. When reading through the discussion, what struck me is that these scarps probably formed in a small number (or possibly one) rupture, as noted by the authors. However, the along-strike patterns of scarp height are interpreted using terminology more

usually associated with multiple-earthquake displacement profiles (e.g. p22, line 21, "...separate faults that have since hard-linked and matured..."). It seems that in a single, or a few, earthquakes, there's not much chance to create new linkages and mature the system, so it would be helpful to discuss how the variations in scarp height might be interpreted on more of a single-event to few-event timescale. From this perspective, the paper could benefit from more discussion of the literature that deals with along-strike variations in how much earthquake slip is expressed on a single scarp, or distributed over a wide area (and therefore very hard to see, or invisible, in the geomorphology). There have been some nice studies of this (e.g. Milliner et al, GRL, doi 10.1002/2016GL069841, 2016; Wang et al, BSSA, doi 10.1785/0120120364, 2014). It would be good to see a discussion of the degree to which the scarp height variations may represent the geomorphological analogues of the variability of on- versus off-fault deformation seen in these recent earthquakes. (These are both strike-slip events, but the same may well be true for normal-faulting – it's worth discussing.)

2. Another point, which in some ways is very closely related to the one above, is the extent to which the scrap height variations may relate to the robustness/weathering of the scarp. Looking at modern normal faulting earthquakes, when we see them in the field the scarps degrade in a very laterally-variable manner over years and decades. In part this relates to whether they are in bedrock or alluvium, but there is also lots of variety within each lithology, possibly related to the degree of fault damage or consolidation of the rocks. A good example are the scarps from the 1981 Corinth earthquake sequence, some of which are still big and dramatic, and some of which have pretty much disappeared. It's hard to say how this might feed into the results of the authors, as there's nothing very quantitative known about the along-strike variations in scarp degradation from the recent events, but it's probably worth some discussion.

3. I liked the synthetic testing, but it would be good to make a couple of small alterations. One is that the examples shown in Fig 4 don't have any profiles that look like the real scarps in Fig 2 (I think because the wavelength and possibly amplitude of

non-scarp topography in the foot-wall isn't big enough). It would be good to see some synthetics that look more like fig 2b. The other thing I thought would be helpful would be to have a graph of the scrap characteristics of the cases where the algorithm failed. The pattern seems to be that the methods and parameters that give the most accurate results are also the ones where there are lots of failures by the algorithm to find scarps. It would therefore be good to know whether there is a systematic bias introduced into the results, based on which type of scarps do and don't get recognized by the algorithm. This discussion of whether any bias is introduced would then be useful for later in the paper, when it comes to interpreting the results from Africa.

4. In terms of the slip-length ratios, it's worth noting that the 2008 Yutian and 2006 Mozambique normal-faulting earthquakes both had a ratios at seismogenic depths of 1-2 x 10ˆ-4. These were mostly blind, but the principle of the biggest continental normal-faulting events we've seen having these ratios suggests that it's not out of the question that the scarps studied in this paper could be due to single events (if they ruptured to the surface). In general, I think this section (and the one about magnitudes) relies too much on the very sparse record of big modern normal-faulting events, either in Africa or elsewhere. They are rare enough (because of the long repeat times) that our modern record is extremely small, and it's an open question how representative it is.

More minor comments:

- There is a general feeling in the paper (mostly in the introduction) that soft-linkage is likely to halt earthquake ruptures. A famous example at Platea-Kaparelli in 1981 involved two fault segments rupturing in the same earthquake, with the deformation between them occurring by spatially-distributed minor normal-faulting, which in the topography would look like soft linkage. This is only one example, and there are others where soft-linkages have halted ruptures, but it would be good to mention somewhere just to keep things balanced.

- p3, line 5 – I think most people actually do this by fitting lines to slopes that are a safe

distance to either side of the scarp, and looking for an offset between them, rather than actually picking the top and bottom of the scarp itself.

- A small inset showing the location within Africa would be helpful for those not familiar with the area.

- P6 L5 – 'the' Pleiades DEM is introduced, but a few notes about its properties/construction would be handy. Also, the resolution is here given as 50 cm, but elsewhere (e.g. Fig 7) as 5 m.

- P6 L15 – a small sketch on one of the figures explaining the geometry described here would be useful.

- P6 L27 (and elsewhere) – I see what is being meant by 'signal-to-noise ratio', but I would describe it as something different (e.g. 'non-tectonic features in the DEM'), to avoid possible confusion with the noise level in the DEM relating to the data and analysis methods used (i.e. the measurement noise).

- P7 L14 – Is it that it 'better represents the average', or that it's using a different definition of 'average' (i.e. median rather than mean)?

- P8, top of page. Most readers probably won't know what G-S and Lowess filters are (including me until I looked it up). It would therefore be good to put some equations and explanation here, to help people see what the filters are actually doing.

- P11 L 17 – there is a mis-match between the labeling of 'moderate' and 'high' between this text and fig 4.

- Fig 4 and 5 – it would be nice to put a big bold line on for 0 misfit, to make it clear which parts of the plots represent the ideal result.

- Fig 10 – it would be good to show the actual values, as well as the moving average.

- Fig 12 – I struggled to see what the splitting of S3 into the three lettered sections was based on, so a clearer explanation would help.

---

## Referee Comment (RC2) · L. Gregory (Referee) · 27 Jun 2018

Summary

This paper describes an automated method of fault scarp detection and measurement. The method is first tested on synthetic data, and then applied to faults in southern Malawi on real digital elevation models with a range of resolution (from 30 m to 50 cm). I think that this tool could be used on a range of normal fault scarps, to make quick automatic maps of displacement. I think this is a nice study with a good level of analysis. However there are some changes that could be made to this manuscript to improve the clarity of the methodology and application to the Malawi faults, and

the limitations of an automatic approach. I think that any reader should be cautious in applying automatic methods to DEMs, and the interpretation of scarp heights must take into account (1) the age of offset features, (2) how variable long-term preservation of fault scarps may be, and (3) how much displacement at the surface reflects the structure and linkage of faults at depth (e.g. how does localised surface slip reflect co-seismic slip at depth. I think that the authors of this paper have done a sufficient job of discussing 2-3 in their study area, but (1) must be explained from the start of the paper. I did not realise until the discussion what timescale the scarps were interpreted to represent, but this is integral to the paper.

As this is an open review, I can state that I agree with and overlap with comments made by R1.

General comments

(1) The introduction needs to give the context of the timescales of displacement calculations. Do the scarps represent a single earthquake, multiple events, or geological offsets? This concept is only introduced in the discussion. I think the introduction should also include a brief description of the climatic setting and local potential for erosion of the scarps – how long should we expect them to persist (e.g. years, 100's, 1000's, kyrs?).

In general, after reading the discussion, it is still not clear to me how the measured scarp heights can be related to fault growth processes that occur over multiple earthquake cycles on geological (e.g. >10 kyr) timescales. Comparing the slip to length for a single event is reasonable, but if these scarps represent multiple earthquakes, then the ratio is not very meaningful unless the displacements represent the total geological offset (e.g. Cowie and Scholz, 1992). I think that the authors try to boil down some very complicated concepts by trying to relate the scarps to both a single event and the long-term geological evolution of the faults. Displacements in a single event can be highly variable along the strike of the fault, and may not always reflect the structure

at depth. This may even be true over a few earthquake cycles. I think there is an interesting question to be addressed, that is, how do persistent variations in surface displacements (over a few EQ's) relate to the structure of the fault?

(2) I like the detailed discussion of the fault evolution and linkage, which is possible due to the high resolution analysis. How old are these faults, and what stage of growth are they in? The authors hint at this in the discussion by suggesting that the Thyolo and Muona faults are more mature than the Malombe faults, but could it just be that the Thyolo and Muona faults have experienced large earthquakes with more shallow slip, more recently?

(3) It would be helpful to include more introduction to how fault scarps are treated in the literature. In other studies, the crest and base of scarps are not used, but instead linear regressions are fit to offset features away from the scarp and projected to the fault. This method avoids problem of near-field scarp degradation and of proximal off-fault deformation (e.g. Nissen et al., 2014; Cowie et al., 2017). It is not clear why the crest-base method has been used in this paper.

(4) I think your algorithm essentially is a way of smoothing out the non-tectonic signals in the data, but then if the scarps are the same size as any non-tectonic signal, they will also be removed (and therefore no scarp will be found in that profile). It would be helpful to the reader to state this more plainly, if it is true? I also wonder if your algorithm is smoothing any off-fault deformation, which may be within 10's of meters of the scarp (e.g. Nissen et al., 2014; Gold et al., 2015).

(5) Overall – addressing the conclusion to the discussion on page 35, I think it is highly likely that the scarps represent multiple events given the extreme slip to length ratios you calculate, but that large magnitude earthquakes are definitely possible (even if they do not rupture the entire fault). Every time there is a new surface-rupturing earthquake, they seem to be even more complex (e.g. New Zealand and C Italy, 2016), so it is not surprising (though very interesting!) that there is some variability of the measured

scarp heights, which may be even more difficult to unravel if they represent several earthquakes but not the total geological history of the faults.

Specific comments

P2, Lines 20-28 hint at the uncertainties associated with DEMS, but these are not discussed formally in the paper. There could be at least one or two references in this paragraph on general DEM processing here or in the methods.

P 7, L 1: Change Media to Median

Pg 8, lines 7-9 – I'm not sure that I consider the profile to represent 'noise' in the data, as the features mentioned to cause noise are real features, which are being reliably recorded in the DEM. I think throughout the manuscript there needs to be a distinction between analytical noise in data and real features in the landscape that may cause ambiguity in the scarp height.

How is the dashed grey line calculated on figure 2a-c? 'By eye' to me, it looks like the manually picked crest of the scarp in 2a is too far to the right on the profile – the scarp looks like it starts at ∼140 m actually closer to the automatic picks.

Figure 4b is nicely summarised in the text on p12, lines 2-6. It would be helpful to do the same for figure 4a, given the amount of work that went into the synthetic tests.

P12, L24-25: Is this analysis in this paper (e.g. do you mean on the real data), or planned future work?

P 18 - It is impressive that scarps with heights less than 5 m can be identified in SRTM! I think this has good implications for using this tool to identify active fault scarps.

Discussion

P29, L18-23: I don't understand the logic behind the choice of average height for the Bilila-Mtakataka fault rupture (11 +/- 7 m). Why are the algorithm results not used? I understand that the rupture surveyed in this paper is not as complete as that from

Hodge et al., 2018? Why not combine the manual picks from the parts of the rupture that were surveyed in the previous work, with the automatic picks from this paper, to have the most data integrated into the average displacement?

P29, L26-27: The 2 km uncertainty seems arbitrary, how was this chosen?

P34, L3: this magnitude calculation is a maximum magnitude – this must be made clear. Smaller magnitude earthquakes can also occur (and also be devastating). I would use the word 'estimated' rather than 'found'.

Figures

Problem with the labels in figure 2 d-f. The text references to the figure do not line up with what is written on the figure – and I cannot tell whether the labels on the figure actually correspond to what is plotted (e.g. in the text (e) is listed as a moving mean, bin width 20 m but plotted as Lowess 40 m – which is correct?). It looks like figure 2f is actually the 40 m bin (Lowess?) because it is smoother. This makes the whole comparison even more confusing for a reader who is not familiar with the different filtering methods.

It would also be helpful to show an example of the Savitzky-Golay filter in figure 2, as it is discussed in section 3 and more readers will not be familiar with this type of filter.

In figure 4, is it possible to also plot the actual fits of the models to the synthetic profile on the first panel – perhaps use the best fitting model, so the reader can get a more physical sense of how well the algorithm is working in the best case?

Figure 6: It would be nice to see an unannotated version of this in the supplementary material, so we can see the scarp. Or even better, use a dashed line or arrows to indicate the trace of the scarp in fig 6. We all like to see a nice fault scarp in a hi-res DEM.

Figure 10: Top of the figure, bold black line is labelled as Manual TanDEM-X 12-m ('Chapter 3') – should it be 'Section 3'?

References:

Biasi and Wesnousky, 2016. Steps and Gaps in ground ruptures: empirical bounds on rupture propagation. Bull Seis. Soc. Am. 106, p 1110-1124. DOI: 10.1785/0120150175

Cowie and Scholz, 1992. Growth of faults by accumulation of seismic slip. JGR 97, no B7, p 11085-11095.

Cowie et al., 2017. Orogen-scale uplift in the central Italian Apennines drives episodic behaviour of earthquake faults. Sci Rep 7:44858, DOI: 10.1038/srep44858

Gold et al., 2015. On and off fault deformation associated with the September 2013 Mw 7.7 Balochistan earthquake: implications for geologic slip rate measurements. Tectonophysics 660, p 65-78, doi: 10.1016/j.tecto.2015.08.019

Nissen et al., 2014. Coseismic fault zone deformation revealed with differential lidar: Examples from Japanese MwâLij7 intraplate earthquakes. EPSL 405, p244-256, DOI: 10.1016/j.epsl.2014.08.031

---

## Author Comment (AC1) · 18 Oct 2018

We thank the reviewers for their constructive reviews and apologize for the delay in returning the manuscript. The lead author, Michael Hodge, has left academia for a job with the Civil Service and the second author, Juliet Biggs, is now acting as corresponding author.

The reviewers raise several interesting points regarding the factors which may affect the along-strike variability of scarp heights. To address these comments, we have added the following paragraph to the end of the introduction:

[Figure]

Our aim is to develop an algorithm capable of measuring along-strike variations in the height of fault scarps at high resolution across a range of settings. The nature of the subsequent analysis and interpretation will, however, depend on the age and type of fault considered as well as the local lithological and climatic conditions. Individual earthquakes can produce scarps of variable height and a mix of on-fault and off-fault deformation (Wang et al., 2014; Gold et al., 2015; Milliner et al., 2016; Nissen et al., 2016). In some circumstances, ruptures are halted by discontinuities or steps in a fault system, whereas other earthquakes produce complex rupture patterns which include multiple fault segments (e.g. Jackson et al., 1982; Hamling et al., 2017). Between earthquakes, erosion depends on variations in lithological and climatic properties, which can produce dramatic changes in scarp height over short distances in only a few decades. For example, some parts of the scarp formed in the 1981 Alkyonides earthquake, Gulf of Corinth, are well-preserved but others have nearly disappeared (e.g. Mechernich et al., 2018). Some fault scarps are formed by individual earthquakes, others are multi-scarps produced by a few events, while others represent the cumulative effects of numerous earthquake cycles over tens of kyrs. In these cases, variations in scarp height may contain information on fault evolution that can be extracted by identifying structural segmentation (e.g. Watterson, 1986; Giba et al., 2012; Manighetti et al., 2015) and the presence of linking structures (e.g. Soliva and Benedicto, 2004; Nicol et al., 2010). However, these long-term eects will be convolved with variations associated with individual earthquakes. This combination of timescales involved in scarp generation raises the question as to what extent variations in oset and erosion persist across multiple earthquake cycles.

Reviewer specific comments.

Reviewer: This paper is an interesting one, which aims to produce an automated method of analyzing DEMs in order to extract metrics about fault scarps. The aim is a good one, in that it will allow scarp geometries to be analyzed with minimal interpretive input, and rapidly over large areas, which will be useful in terms of documenting

and interpreting any along-strike variations that are present (e.g. due to segmentation). I have some comments that I hope will help to improve the clarity of the manuscript.

1. When reading through the discussion, what struck me is that these scarps probably formed in a small number (or possibly one) rupture, as noted by the authors. However, the along-strike patterns of scarp height are interpreted using terminology more usually associated with multiple-earthquake displacement profiles (e.g. p22, line 21, . . .separate faults that have since hard-linked and matured. . .). It seems that in a single, or a few, earthquakes, theres not much chance to create new linkages and mature the system, so it would be helpful to discuss how the variations in scarp height might be interpreted on more of a single-event to few-event timescale. From this perspective, the paper could benefit from more discussion of the literature that deals with along-strike variations in how much earthquake slip is expressed on a single scarp, or distributed over a wide area (and therefore very hard to see, or invisible, in the geomorphology). There have been some nice studies of this (e.g. Milliner et al, GRL, doi 10.1002/2016GL069841, 2016; Wang et al, BSSA, doi 10.1785/0120120364, 2014). It would be good to see a discussion of the degree to which the scarp height variations may represent the geomorphological analogues of the variability of on- versus off-fault deformation seen in these recent earthquakes. (These are both strike-slip events, but the same may well be true for normal-faulting its worth discussing.)

Response: We agree with this point. The issue of off-fault deformation has now be raised in the introduction to acknowledge, up front, the problem of distinguishing whether along-strike variation in scarp morphology relates to cumulative effect of multiple, segmented, earthquakes, or slip variation within a single rupture. We also point out the possibility of scarp height variations reflecting single earthquake complexity in the Discussion Section 7.2 (p. 30, lines 18-19 and p.32 lines 30-35) and Conclusion (p. 36, line 34 onwards).

2. Another point, which in some ways is very closely related to the one above, is the extent to which the scrap height variations may relate to the robustness/weathering of

the scarp. Looking at modern normal faulting earthquakes, when we see them in the field the scarps degrade in a very laterally-variable manner over years and decades. In part this relates to whether they are in bedrock or alluvium, but there is also lots of variety within each lithology, possibly related to the degree of fault damage or consolidation of the rocks. A good example are the scarps from the 1981 Corinth earthquake sequence, some of which are still big and dramatic, and some of which have pretty much disappeared. Its hard to say how this might feed into the results of the authors, as theres nothing very quantitative known about the along-strike variations in scarp degradation from the recent events, but its probably worth some discussion.

Response: We agree with this point. The issue of variable erosion has now be raised in the introduction and the variable preservation of the 1981 Corinth earthquake scarp has been used as an illustrative example.

3. I liked the synthetic testing, but it would be good to make a couple of small alterations. One is that the examples shown in Fig 4 dont have any profiles that look like the real scarps in Fig 2 (I think because the wavelength and possibly amplitude of non-scarp topography in the foot-wall isnt big enough). It would be good to see some synthetics that look more like Fig 2b.

Response: The profiles are intended to represent scarps from around the world as we want to ensure the algorithm is sufficiently generic and avoid over-tuning the methods to the specific cases shown in Figure 2 and subsequently. The algorithm is later re-tuned to the specific characteristics of the Malawian through selection of the most appropriate bin width and threshold using a small number of manually analysed examples which are more similar to those seen in Figure 2.

3. The other thing I thought would be helpful would be to have a graph of the scrap characteristics of the cases where the algorithm failed. The pattern seems to be that the methods and parameters that give the most accurate results are also the ones where there are lots of failures by the algorithm to

find scarps. It would therefore be good to know whether there is a systematic bias introduced into the results, based on which type of scarps do and dont get recognized by the algorithm. This discussion of whether any bias is introduced would then be useful for later in the paper, when it comes to interpreting the results from Africa.

Response: This is a common feature of automated algorithms - the most accurate methods are often only applicable to a very small number of 'ideal' cases, whereas looser methods may be applicable in a wider number of cases at the cost of accuracy. Investigating the bias this introduces is an interesting idea and in an ideal world, we would investigate it further. However, with the lead author no longer available, a thorough investigation is beyond the scope of achievable revisions. Instead we have added the following sentence to the discussion of algorithm performance 'It is possible that the selection of scarps biases the analysis of scarp height. However, any bias would be towards the larger, sharper scarps and the effect is likely to be minor in comparison to the effects of erosion which tend to reduce estimates of scarp height.'

4. In terms of the slip-length ratios, its worth noting that the 2008 Yutian and 2006 Mozambique normal-faulting earthquakes both had a ratios at seismogenic depths of 1-2 x 10-4. These were mostly blind, but the principle of the biggest continental normal faulting events weve seen having these ratios suggests that its not out of the question that the scarps studied in this paper could be due to single events (if they ruptured to the surface). In general, I think this section (and the one about magnitudes) relies too much on the very sparse record of big modern normal-faulting events, either in Africa or elsewhere. They are rare enough (because of the long repeat times) that our modern record is extremely small, and its an open question how representative it is.

Response: Thanks! We have added the following sentence: The slip-to-length ratios for the normal-faulting 2008 Yutian and 2006 Mozambique earthquakes were 1-2 x 10-4 although both were blind earthquakes which did not rupture the surface (Elliott et al., 2010; Copley et al., 2012). We believe that the present revision of this section suitably reviews cases that set the precedent of plausible coseismic slip to length ratios without

too heavily prescribing expectations based on limited observations.

Minor comments:

1. There is a general feeling in the paper (mostly in the introduction) that soft-linkage is likely to halt earthquake ruptures. A famous example at Platea-Kaparelli in 1981 involved two fault segments rupturing in the same earthquake, with the deformation between them occurring by spatially-distributed minor normal-faulting, which in the topography would look like soft linkage. This is only one example, and there are others where soft-linkages have halted ruptures, but it would be good to mention somewhere just to keep things balanced.

Response: Good point. We have inserted a comment in the introduction which says: 'In some circumstances, ruptures are halted by discontinuities or steps in a fault system, whereas other earthquakes produce complex rupture patterns which include multiple fault segments (e.g. Jackson et al., 1982; Hamling et al., 2017)'

2. p3, line 5 I think most people actually do this by fitting lines to slopes that are a safe distance to either side of the scarp, and looking for an offset between them, rather than actually picking the top and bottom of the scarp itself.

Response: Apologies, we explained our methodology poorly in the previous version of the manuscript. While we do pick the location of the crest and base of the scarp to measure scarp width, the height is measured by fitting regression lines as suggested by the reviewer. We have made this clearer by rewording the method description (p. 3, line 5; p. 7, paragraph from line 8 onwards).

3. A small inset showing the location within Africa would be helpful for those not familiar with the area.

Response: Done, included on Fig 1 now.

4. P6 L5 the Pleiades DEM is introduced, but a few notes about its properties/construction would be handy. Also, the resolution is here given as 50 cm, but

elsewhere (e.g. Fig 7) as 5 m.

Response: We have amended references to DEM resolution throughout; it is indeed 5 m. 50 cm is the nominal resolution of the original satellite imagery, but to construct a robust DEM of thorough spatial coverage, we downsample the calculated topographic point cloud.

5. P6 L15 a small sketch on one of the figures explaining the geometry described here would be useful.

Response: We have added a reference to Figure 3 where this geometry is shown

6. P6 L27 (and elsewhere) I see what is being meant by signal-to-noise ratio, but I would describe it as something different (e.g. non-tectonic features in the DEM), to avoid possible confusion with the noise level in the DEM relating to the data and analysis methods used (i.e. the measurement noise).

Response: This is a good point. Throughout the paper, we have replaced 'signal-to-noise ratio' with better description of what we mean, typically 'non-tectonic features' or something to that effect.

7. P7 L14 Is it that it better represents the average, or that its using a different definition of average (i.e. median rather than mean)?

Response: We have adjusted the text here slightly to indicate that the median is preferred as a representation of the average, rather than that we calculated these two values separately.

8. P8, top of page. Most readers probably wont know what G-S and Lowess filters are (including me until I looked it up). It would therefore be good to put some equations and explanation here, to help people see what the filters are actually doing. We understand that many of the Solid Earth readership will not be familiar with these filters, but references are provided to a wide literature.

Response: These are standard filters which are simple to implement in matlab or python. However, they cannot be represented by a single equation, and a full explanation seems inappropriate for this journal when there is a wide literature already available.

9. P11 L 17 there is a mis-match between the labeling of moderate and high between this text and Fig 4.

Response: The terminology has been updated (see previous comments) and this is no longer an issue

10. Fig 4 and 5 it would be nice to put a big bold line on for 0 misfit, to make it clear which parts of the plots represent the ideal result.

Response: After consideration, we have not made this alteration. The line could be misconstrued as the ideal parameters, whereas in fact they're just the result of extrapolations between two values either side of 0

11. Fig 10 it would be good to show the actual values, as well as the moving average.

Response: Done, included in Fig 10 now.

12. Fig 12 I struggled to see what the splitting of S3 into the three lettered sections was based on, so a clearer explanation would help.

Response: This is clearest in the map view, so we have changed the figure reference to Fig 12d to help the reader identify the key points.

References: 1. Biasi and Wesnousky, 2016. Steps and Gaps in ground ruptures: empirical bounds on rupture propagation. Bull Seis. Soc. Am. 106, p 1110-1124. DOI:10.1785/0120150175 2. Cowie and Scholz, 1992. Growth of faults by accumulation of seismic slip. JGR 97, no B7, p 11085-11095. 3. Cowie et al., 2017. Orogen-scale uplift in the central Italian Apennines drives episodic behaviour of earthquake faults. Sci Rep 7:44858, DOI: 10.1038/srep44858 4. Gold et al., 2015. On

and off fault deformation associated with the September 2013 Mw 7.7 Balochistan earthquake: implications for geologic slip rate measurements. Tectonophysics 660, p 65-78, doi: 10.1016/j.tecto.2015.08.019 5. Nissen et al., 2014. Coseismic fault zone deformation revealed with differential lidar: Examples from Japanese MwLij7 intraplate earthquakes. EPSL 405, p244-256, DOI: 10.1016/j.epsl.2014.08.031
* * *
[Figure]

**Fig. 1.** Fig 1 with new EARS inset

Lowess (20 m)

Moving Mean (20 m)

Savitzky-Golay (20 m)

Lowess (40 m)

Elevation (m)

Slope θ (°) / Derivative of Slope (°/m)

Distance (m)

H 22 m, W 45 m
ā 26°

H 18 m, W 35 m
ā 27°

H 16 m, W 20 m
ā 39°

**Fig. 2.** Fig 2 with correct labels and S-G smoothing example

**Fig. 3.** Fig 4 with scarp markers

[Figure]

**Bilila-Mtakataka Fault**

Algorithm Moving Mean (1 km) • Datapoint
Algorithm Standard Devation (1 km)
Manual TanDEM-X 12 m (Hodge et al. 18, *GRL*)
Breaks in scarp continuity (over 1 km)

**Fig. 4.** Fig 10 with raw points

---

## Author Comment (AC2) · 18 Oct 2018

We thank the reviewers for their constructive reviews and apologize for the delay in returning the manuscript. The lead author, Michael Hodge, has left academia for a job with the Civil Service and the second author, Juliet Biggs, is now acting as corresponding author.

The reviewers raise several interesting points regarding the factors which may affect the along-strike variability of scarp heights. To address these comments, we have added the following paragraph to the end of the introduction:

[Figure]

Our aim is to develop an algorithm capable of measuring along-strike variations in the height of fault scarps at high resolution across a range of settings. The nature of the subsequent analysis and interpretation will, however, depend on the age and type of fault considered as well as the local lithological and climatic conditions. Individual earthquakes can produce scarps of variable height and a mix of on-fault and off-fault deformation (Wang et al., 2014; Gold et al., 2015; Milliner et al., 2016; Nissen et al., 2016). In some circumstances, ruptures are halted by discontinuities or steps in a fault system, whereas other earthquakes produce complex rupture patterns which include multiple fault segments (e.g. Jackson et al., 1982; Hamling et al., 2017). Between earthquakes, erosion depends on variations in lithological and climatic properties, which can produce dramatic changes in scarp height over short distances in only a few decades. For example, some parts of the scarp formed in the 1981 Alkyonides earthquake, Gulf of Corinth, are well-preserved but others have nearly disappeared (e.g. Mechernich et al., 2018). Some fault scarps are formed by individual earthquakes, others are multi-scarps produced by a few events, while others represent the cumulative effects of numerous earthquake cycles over tens of kyrs. In these cases, variations in scarp height may contain information on fault evolution that can be extracted by identifying structural segmentation (e.g. Watterson, 1986; Giba et al., 2012; Manighetti et al., 2015) and the presence of linking structures (e.g. Soliva and Benedicto, 2004; Nicol et al., 2010). However, these long-term effects will be convolved with variations associated with individual earthquakes. This combination of timescales involved in scarp generation raises the question as to what extent variations in offset and erosion persist across multiple earthquake cycles.

Reviewer specific comments.

This paper describes an automated method of fault scarp detection and measurement. The method is first tested on synthetic data, and then applied to faults in southern Malawi on real digital elevation models with a range of resolution (from 30 m to 50 cm). I think that this tool could be used on a range of normal fault scarps, to make

quick automatic maps of displacement. I think this is a nice study with a good level of analysis. However there are some changes that could be made to this manuscript to improve the clarity of the methodology and application to the Malawi faults, and the limitations of an automatic approach. I think that any reader should be cautious in applying automatic methods to DEMs, and the interpretation of scarp heights must take into account (1) the age of offset features, (2) how variable long-term preservation of fault scarps may be, and (3) how much displacement at the surface reflects the structure and linkage of faults at depth (e.g. how does localised surface slip reflect co-seismic slip at depth. I think that the authors of this paper have done a sufficient job of discussing 2-3 in their study area, but (1) must be explained from the start of the paper. I did not realise until the discussion what timescale the scarps were interpreted to represent, but this is integral to the paper.

As this is an open review, I can state that I agree with and overlap with comments made by R1.

General comments:

(1) The introduction needs to give the context of the timescales of displacement calculations. Do the scarps represent a single earthquake, multiple events, or geological offsets? This concept is only introduced in the discussion. I think the introduction should also include a brief description of the climatic setting and local potential for erosion of the scarps how long should we expect them to persist (e.g. years, 100s, 1000s, kyrs?). In general, after reading the discussion, it is still not clear to me how the measured scarp heights can be related to fault growth processes that occur over multiple earthquake cycles on geological (e.g. >10 kyr) timescales. Comparing the slip to length for a single event is reasonable, but if these scarps represent multiple earthquakes, then the ratio is not very meaningful unless the displacements represent the total geological oset (e.g. Cowie and Scholz, 1992). I think that the authors try to boil down some very complicated concepts by trying to relate the scarps to both a single event and the long-term geological evolution of the faults. Displacements in a single

event can be highly variable along the strike of the fault, and may not always reflect the structure at depth. This may even be true over a few earthquake cycles. I think there is an interesting question to be addressed, that is, how do persistent variations in surface displacements (over a few EQs) relate to the structure of the fault?

Response: The SPARTA algorithm is designed to measure surface offset regardless of whether the scarp formed during a single or multiple events, whereas the interpretation and analysis of the results would differ significantly. In order to better set up this problem, we have added to both the introduction and discussion sections:

1) We have added the following sentences to the first paragraph of the introduction to introduce the range of possible scarp types that the algorithm could be used to study 'The scarps may be formed by a single earthquake or a small number of events, or represent the cumulative effect of numerous events over geological timescales. Linking the surface offset along the fault to information on the age of the features can provide information about the rupture and slip history on the fault (e.g., Wallace, 1968; Sieh, 1978; Zielke et al., 2012; Ren et al., 2016). For mature faults, it can be used to characterise long-term development by identifying structural segmentation (e.g., Watterson, 1986; Giba et al., 2012; Manighetti et al., 2015) and the presence of linking structures (e.g., Soliva and Benedicto, 2004; Nicol et al., 2010).'

2) To provide more background information on Malawi examples, we have added the following to the start of section 2 (2nd paragraph) ' In the southern MRS, the Bilila-Mtakataka fault (BMF) scarp breaks the surface along almost its entire length, a distance of 110 km (Jackson and Blenkinsop, 1997). Previous studies suggested that the scarp formed during a single earthquake (Jackson and Blenkinsop, 1997), but the morphology and geometry vary along strike (Hodge et al., 2018a) and are more typical of a large, structurally segmented normal fault which has experienced several previous earthquake cycles. (e.g., Schwartz and Coppersmith, 1984; Wesnousky, 1986; Peacock and Sanderson, 1991)

We agree with the reviewer that slip-to-length analyses are only meaningful when applied to individual earthquakes Our analysis of slip-to-length ratio is a test of the hypothesis that the scarps were formed by a single event and clearly states the assumptions made. The unusual slip-to-length ratios as then used as another line of evidence that multiple earthquakes were responsible. We have inserted some comments throughout the text to make this clearer, by explaining that assuming a single earthquake is an end-member, and by more explicitly interpreting the high slip/length ratios as indicating that the scarps likely formed by multiple earthquakes.

(2) I like the detailed discussion of the fault evolution and linkage, which is possible due to the high resolution analysis. How old are these faults, and what stage of growth are they in? The authors hint at this in the discussion by suggesting that the Thyolo and Muona faults are more mature than the Malombe faults, but could it just be that the Thyolo and Muona faults have experienced large earthquakes with more shallow slip, more recently?

Response: Little data are available on the faults in southern Malawi, which is part of the argument to study their geomorphology. We do, however, note that the Malombe fault is an intra-basin fault, whereas the other faults are in the immediate footwall of rift border escarpments (p. 5, lines 19-22). For balance, we also add a statement at the end of the discussion on fault evolution (end of section 7.2.3) to say that: In the absence of any dated earthquakes, however, we cannot rule out that the Thyolo and Muona faults have experienced a more recent earthquake, with surface rupture, than the Malombe and Bilila-Mtakataka faults, and therefore have the appearance of being more mature.

(3) It would be helpful to include more introduction to how fault scarps are treated in the literature. In other studies, the crest and base of scarps are not used, but instead linear regressions are fitted to offset features away from the scarp and projected to the fault. This method avoids problem of near-field scarp degradation and of proximal off fault deformation (e.g. Nissen et al., 2014; Cowie et al., 2017). It is not clear why the

crest-base method has been used in this paper.

Response: As explained above to a similar comment from Reviewer 1, we did use regressions fitted to the offset surfaces either side of the scarp to calculate scarp height. We have now made this clearer, and apologise for confusion caused by the wording of the original methods description.

(4) I think your algorithm essentially is a way of smoothing out the non-tectonic signals in the data, but then if the scarps are the same size as any non-tectonic signal, they will also be removed (and therefore no scarp will be found in that profile). It would be helpful to the reader to state this more plainly, if it is true?

Response: We have added a comment to this effect to the discussion (first paragraph,section 7.1)

(4 cont) I also wonder if your algorithm is smoothing any off-fault deformation, which may be within 10s of meters of the scarp (e.g. Nissen et al., 2014; Gold et al., 2015).

Response: For discussion of off-fault deformation, see the response to reviewer 1, point 1

(5) Overall addressing the conclusion to the discussion on page 35, I think it is highly likely that the scarps represent multiple events given the extreme slip to length ratios you calculate, but that large magnitude earthquakes are definitely possible (even if they do not rupture the entire fault). Every time there is a new surface-rupturing earthquake, they seem to be even more complex (e.g. New Zealand and C Italy, 2016), so it is not surprising (though very interesting!) that there is some variability of the measured scarp heights, which may be even more difficult to unravel if they represent several earthquakes but not the total geological history of the faults.

Response: We think this comment is covered by the additional introductory paragraph added in response to comments from both reviewers, and by an additional statement near the end of the Conclusions which says: Our results suggest that each fault has

likely formed through multiple events. However, earthquake ruptures are known to show complex variation in on- and off-fault deformation along-strike, and it is possible that along-strike variations in scarp height also reflect near-surface slip distribution in single earthquakes. It is also important to note that even if the current scarps formed in multiple events, large magnitude (M 7 - 8) earthquakes are possible in Malawi.

Specific comments:

(1) P2, Lines 20-28 hint at the uncertainties associated with DEMS, but these are no discussed formally in the paper. There could be at least one or two references in this paragraph on general DEM processing here or in the methods.

Response: We have added a sentence indicating the errors found in comparison of stereo-image DEMs with lidar 'ground truth', which essentially captures noise in the data (Citing Zhou 2015).

(2) P 7, L 1: Change Media to Median

Response: Thanks for the catch!

(3) Pg 8, lines 7-9 Im not sure that I consider the profile to represent noise in the data, as the features mentioned to cause noise are real features, which are being reliably recorded in the DEM. I think throughout the manuscript there needs to be a distinction between analytical noise in data and real features in the landscape that may cause ambiguity in the scarp height.

Response: See response to Reviewer 1, we have replaced references to 'noise' with more specific reference to topographic, natural features that make identifying the scarp in the DEM difficult and/or ambiguous.

(4) How is the dashed grey line calculated on figure 2a-c? By eye to me, it looks like the manually picked crest of the scarp in 2a is too far to the right on the profile the scarp looks like it starts at 140 m actually closer to the automatic picks.

[Figure]

Response: The main motivation for this algorithm and paper is that scarp location is open to interpretation, and so requires some unified, quantified, repeatable metrics. Investigators' choice of location may almost always be biased because of this selection of metrics-our algorithm makes this process quantitative and repeatable. We note that the reviewer's contention of our pick illustrates the issue we are addressing.

(5) Figure 4b is nicely summarised in the text on p12, lines 2-6. It would be helpful to do the same for Figure 4a, given the amount of work that went into the synthetic tests.

Response: Fig 4a is summarised in the previous paragraph (p11, lines 6-15 of the submitted version). While we appreciate the reviewer acknowledging the time spent generating these profiles describing each profile-filter combination in detail is rather repetitive. In fact, I think we removed this from an earlier draft of the manuscript in the interest of maintaining the readers attention. A really attentive reader could easily extract the corresponding information from Fig 4a given the description provided of Fig 4

(6) P12, L24-25: Is this analysis in this paper (e.g. do you mean on the real data), or planned future work?

Response: This is something that has been done, and we have rephrased accordingly 'Thus these are the optimal filters which we choose to employ in our natural measurements, using bin width and slope thresholds tailored to the local environment'

(7) P 18 - It is impressive that scarps with heights less than 5 m can be identified in SRTM!

Response: Yes, we agree. It can mainly be attributed to this ideal setting with low-relief, relatively level landscapes surrounding these faults. Naturally in higher relief areas such small scarps may not be evident in 30-m SRTM data.

(8) I think this has good implications for using this tool to identify active fault scarps.

Response: Absolutely, and to quantify them!

Discussion:

(1) P29, L18-23: I dont understand the logic behind the choice of average height for the Bilila-Mtakataka fault rupture (11 +/- 7 m). Why are the algorithm results not used? I understand that the rupture surveyed in this paper is not as complete as that from Hodge et al., 2018? Why not combine the manual picks from the parts of the rupture that were surveyed in the previous work, with the automatic picks from this paper, to have the most data integrated into the average displacement?

Response: This is a valid point. We have combined the findings from Hodge et al 2018 from 0 - 8 km (7 +/- 3 m) and 98 - 110 km (10 +/- 6 m), and the findings from this paper to give a measurement of 16 +/- 7 m for the entire scarp. This is a weighted average. The calculations in the paper have been adjusted accordingly.

(2) P29, L26-27: The 2 km uncertainty seems arbitrary, how was this chosen?

Response: Changed to 1% and updated calculations accordingly.

(3) P34, L3: this magnitude calculation is a maximum magnitude this must be made clear. Smaller magnitude earthquakes can also occur (and also be devastating). I would use the word estimated rather than found.

Response: We have adjusted language in this section to indicate that these are estimates of the maximum plausible magnitude of an earthquake that ruptures the entire fault plane. We have also adjusted the reported magnitudes of several historical quakes listed and now use the modern reanalysis of moments which are often smaller

Figures:

(1) Problem with the labels in Figure 2 d-f. The text references to the figure do not line up with what is written on the figure and I cannot tell whether the labels on the figure actually correspond to what is plotted (e.g. in the text (e) is listed as a moving mean, bin width 20 m but plotted as Lowess 40 m which is correct?). It looks like Figure 2f is actually the 40 m bin(Lowess?) because it is smoother. This makes the whole

comparison even more confusing for a reader who is not familiar with the different filtering methods.

Response: Done

(2) It would also be helpful to show an example of the Savitzky-Golay filter in Figure 2, as it is discussed in section 3 and more readers will not be familiar with this type of filter.

Response: Done

(3) In Figure 4, is it possible to also plot the actual fits of the models to the synthetic profile on the first panel perhaps use the best fitting model, so the reader can get a more physical sense of how well the algorithm is working in the best case?

Response: Done

(4) Figure 6: It would be nice to see an unannotated version of this in the supplementary material, so we can see the scarp. Or even better, use a dashed line or arrows to indicate the trace of the scarp in Fig 6. We all like to see a nice fault scarp in a hi-res DEM.

Response: Done

(5) Figure 10: Top of the figure, bold black line is labelled as Manual TanDEM-X 12-m (Chapter 3) should it be Section 3?

Response: Yes, Corrected
* * *
**Fig. 1.** Fig 1 with EARS inset

**Fig. 2.** Fig 2 with S-G filter example

**Fig. 3.** Fig 4 with scarp markers

[Figure]

**Fig. 4.** Fig 10 with raw points